# Upper-ocean temperature characteristics in the subantarctic southeastern Pacific based on biomarker reconstructions

**Julia Rieke Hagemann[1], Lester Lembke-Jene[1], Frank Lamy[1], Maria-Elena Vorrath[2], Jérôme Kaiser[3], Juliane Müller[1], Helge W. Arz[3], Jens Hefter[1], Andrea Jaeschke[4], Nicoletta Ruggieri[1], and Ralf Tiedemann[1]**

[1]Alfred Wegener Institute, Helmholtz Centre for Polar and Marine Research, 27570 Bremerhaven, Germany

[2]Institute for Geology, University of Hamburg, 20146 Hamburg, Germany

[3]Leibniz Institute for Baltic Sea Research Warnemünde, 18119 Rostock, Germany

[4]Institute of Geology and Mineralogy, University of Cologne, 50923 Cologne, Germany

**Correspondence:** Julia Rieke Hagemann (julia.hagemann@awi.de) and Lester Lembke-Jene (lester.lembke-jene@awi.de)

**Abstract.** As remnants of living organisms, alkenones and isoprenoid glycerol dialkyl glycerol tetraether lipids (isoGDGTs) are widely used biomarkers for determining ocean water temperatures from the past. The organisms that these proxy carriers stem from are influenced by a number of environmental parameters, such as water depth, nutrient availability, light conditions, or seasonality, which all may significantly bias the calibration to ambient water temperatures. Reliable temperature determinations thus remain challenging, especially in higher latitudes and for undersampled regions. We analyzed 33 sediment surface samples from the southern Chilean continental margin and the Drake Passage for alkenones and isoGDGTs and compared the results with gridded instrumental reference data from the World Ocean Atlas 2005 (WOA05) and previously published data from an extended study area covering the central and western South Pacific towards the Aotearoa / New Zealand continental margin. We show that for alkenone-derived sea surface temperatures (SSTs), the widely used global core-top calibration of Müller et al. (1998) yields the smallest deviation of the WOA05-based SSTs. On the contrary, the calibration of Sikes et al. (1997), determined for higher latitudes and summer SSTs, overestimates modern WOA05-based SSTs in both the annual mean and summer. Our alkenone SSTs show a slight seasonal shift of $\sim 1\,^\circ$C at the southern Chilean margin and up to $\sim 2\,^\circ$C in the Drake Passage towards austral summer SSTs. Samples in the central South Pacific, on the other hand, reflect an annual mean signal. We show that for isoGDGT-based temperatures, the subsurface calibration of Kim et al. (2012a) best reflects temperatures from the WOA05 in areas north of the Subantarctic Front (SAF). Temperatures south of the SAF are, in contrast, significantly overestimated by up to $14\,^\circ$C, irrespective of the applied calibration. In addition, we used the GDGT [2]/[3] ratios, which give an indication of the production depth of the isoGDGTs and/or potential influences from land. Our samples reflect a subsurface (0 to 200 m water depth) rather than a surface (0–50 m water depth) signal in the entire study area and show a correlation with the monthly dust distribution in the South Pacific, indicating terrigenous influences. The overestimation of isoGDGT surface and subsurface temperatures south of the SAF highlights the need for a reassessment of existing calibrations in the polar Southern Ocean. Therefore, we suggest a modified Southern Ocean tetraether index (TEX$_{86}$)-based calibration for surface and subsurface temperatures, which shows a lower temperature sensitivity and yields principally lower absolute temperatures, which align more closely with WOA05-derived values and also OH–isoGDGT-derived temperatures.

## 1 Introduction

Alkenones (e.g., Brassell et al., 1986; Herbert, 2001, 2014) and isoGDGTs (isoprenoid glycerol dialkyl glycerol tetraether lipids; Schouten et al., 2002, 2013a) are widely used for determining ocean water temperatures from the past. These biomarkers are present in all oceans and occur from

the tropics to high latitudes (e.g., Herbert et al., 2010; Sikes et al., 1997; Müller et al., 1998; Conte et al., 2006). Alkenone-derived sea surface temperatures (SSTs) are based on the lipid remains of photoautotrophic coccolithophorids (e.g., Baumann et al., 2005; Brassell et al., 1986). The ratio of di- and tri-unsaturated alkenones, expressed as the unsaturation ketone index $U_{37}^{K'}$ (Table A1), reflects SSTs (Prahl and Wakeham, 1987). The calculation of SSTs is based on calibration equations developed over the past $\sim 40$ years (Table A1). Most of these empirically derived equations are relatively similar and based on a comparison of either culture experiments (Prahl et al., 1988; Prahl and Wakeham, 1987) or surface sediment samples from tropical to subpolar regions, with corresponding instrumental data (Müller et al., 1998). Other calibrations (e.g., Sikes et al., 1997) were developed specifically for (sub)polar regions and are adapted for a seasonal bias toward summer SSTs. Henceforth, we use the terms Müller98 for the calibration by Müller et al. (1998) and Sikes97 for the calibration by Sikes et al. (1997; Table A1).

The accuracy of alkenone-based calibrations can be influenced by other environmental factors besides temperature, such as light levels, changes in growth rate, or nutrient availability, but none of these factors seems to have an appreciable effect on the $U_{37}^{K'}$ index (e.g., Caniupán et al., 2014; Epstein et al., 2001; Herbert, 2001; Müller et al., 1998; Popp et al., 1998). In contrast, the preferential degradation of the alkenone $C_{37:3}$ in sediment under aerobic conditions may bias the $U_{37}^{K'}$ signal towards warmer SSTs (Prahl et al., 2010). Seasonality often plays a significant role at high latitudes (e.g., Max et al., 2020; Prahl et al., 2010), due to the primary production being more pronounced in the thermal summer season and annual temperature differences that increase with increasing latitude. In our study region, samples from the central South Pacific most likely represent either summer temperatures (with the Sikes97 calibration) or an annual mean (Müller98 calibration; Jaeschke et al., 2017). Prahl et al. (2010), using samples from the Chilean continental slope, found a slight seasonal summer bias south of $\sim 50°$ S. In contrast, studies from the North Pacific show a seasonal signal towards late summer to autumn SSTs that differ from the annual mean by up to $6°$C (e.g., Max et al., 2020; Prahl et al., 2010).

IsoGDGTs are lipid remains of *Thaumarchaeota* (formerly called *Crearchaeota*, Group I; Brochier-Armanet et al., 2008) that include a certain number of moieties, which increases with growth temperature (Schouten et al., 2002). The lipids glycerol dialkyl glycerol tetraether (GDGT)-0, GDGT-1, GDGT-2, and GDGT-3 contain zero to three cyclopentane moieties in their molecule structure, whereas crenarchaeol and its isomer cren' feature four cyclopentane and one hexane moieties. These ring structures regulate membrane fluidity of *Thaumarchaeota* and change as an adaption to their ambient temperature (Chong, 2010; Gabriel and Chong, 2000; Schouten et al., 2002). The determination of isoGDGT-derived water temperatures is based on the tetraether index

(TEX$_{86}$; Schouten et al., 2002), or its modifications TEX$_{86}^{H}$ and TEX$_{86}^{L}$ (Table A1), which have been determined for water temperatures of $> 15°$C and $< 15°$C, respectively (Kim et al., 2010). While TEX$_{86}^{H}$ is a logarithmic function of the original index, TEX$_{86}^{L}$ omits the GDGT-3 from the denominator and removes the isomer cren' from the equation, due to a weaker correlation to water temperatures in cold regions (Kim et al., 2010). In addition to cyclopentane moieties, three OH–isoGDGTs may also contribute to ambient an temperature adaption (OH–isoGDGT-0, OH–isoGDGT-1, and OH–isoGDGT-2). These OH–isoGDGTs occur globally but in higher amounts in the polar regions, as a further adaption of the *Thaumarchaeota* to cold environments (Fietz et al., 2013; Huguet et al., 2013; Liu et al., 2020). OH–isoGDGTs are frequently so important in polar regions or during glacial phases that it has been recommended to include them in temperature calibrations (Fietz et al., 2016, 2020). In contrast to photo-autotrophic coccolithophores, *Thaumarchaeota* occur throughout the water column (Karner et al., 2001), which complicates the attribution of the reconstructed temperature signal to specific water depths. In general, it is assumed that *Thaumarchaeota* predominantly reflect either a subsurface, i.e., seasonal mixed-layer temperature ($T_{sub}$; 0–200 m water depth), or an SST signal (Table A1) because the grazing and repacking of isoGDGTs into fecal pellets occurs most effectively within the photic zone (Wuchter et al., 2005). The GDGT [2]/[3] ratio can be used to roughly determine the habitat depth of the *Thaumarchaeota*, since it increases with increasing water depth (Dong et al., 2019; Hernández-Sánchez et al., 2014; Kim et al., 2015, 2016; Schouten et al., 2012; Taylor et al., 2013). For the subpolar and polar Southern Ocean, in particular the extensive SE Pacific sector, to date only a little bit of information exists about the applicability of these different temperature proxies and their respective calibrations. In addition, systematic comparisons between alkenone and isoGDGT-based temperature reconstructions using surface sediments have thus far been limited (e.g., Jaeschke et al., 2017; Kaiser et al., 2015). We use the terms SST$^{H}$Kim and SST$^{L}$Kim for the two global surface calibrations by Kim et al. (2010), $T_{sub}^{H}$Kim and $T_{sub}^{L}$Kim for the two global subsurface calibrations by Kim et al. (2012a, b), and SST$^{H}$Kaiser and $T_{sub}^{H}$Kaiser for the local surface and subsurface calibration by Kaiser et al. (2015; Table A1).

In this study, we present a new set of 33 sediment surface samples located along the southern Chilean margin (SCM) and the Drake Passage (DP; $\sim 52$–$62°$ S) to determine upper-ocean water temperatures based on alkenones ($U_{37}^{K'}$) and isoGDGTs (TEX$_{86}^{H}$ and TEX$_{86}^{L}$). We compare our regional results with previously published data from an extended temperate to subpolar South Pacific study area (Fig. 1).

We assess the applicability of the Müller98 and Sikes97 calibrations with World Ocean Atlas (WOA05)-based temperatures and investigate the influence of seasonality on alkenone-based temperature reconstructions. Furthermore, we compare the isoGDGT-based indices TEX$_{86}^{H}$ and TEX$_{86}^{L}$

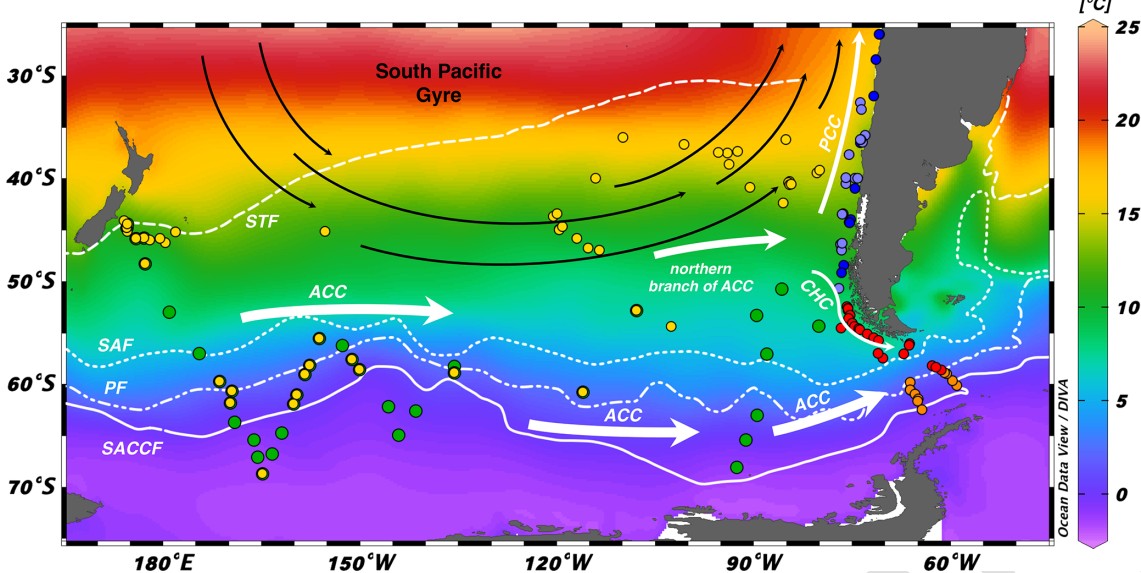

**Figure 1.** Map with SSTs (WOA05; Locarnini et al., 2006) of the extended study area and sample locations. ACC is the Antarctic Circumpolar Current; PCC is the Peru–Chile Current; CHC is the Cape Horn Current; STF is the subtropical front; SAF is the Subantarctic Front; PF is the polar front; SACCF is the southern Antarctic Circumpolar Current front. Red dots indicate the southern Chilean margin and Drake Passage samples (this study); orange dots indicate the Drake Passage samples (Lamping et al., 2021; this study); light blue dots indicate the northern–central Chilean margin samples (Prahl et al., 2006, 2010); dark blue dots indicate the northern–central Chilean margin samples (Kaiser et al., 2015); yellow dots indicate the South Pacific Gyre, central South Pacific, and Aotearoa / New Zealand margin samples (Jaeschke et al., 2017); and green dots indicate the central South Pacific and Aotearoa / New Zealand margin samples (Ho et al., 2014).

and their most common calibrations for SST and $T_{sub}$ (Table A1) with WOA05-based temperatures. Last, we check the potential influence of habitat depth on signal incorporation on the basis of the GDGT [2]/[3] ratio and propose a new calibration specifically for the polar Pacific sector of the Southern Ocean (SO) south of the Subantarctic Front.

## 2   Study area

Our study area comprises the subpolar and polar SE Pacific sector of the Southern Ocean (SO), including the Drake Passage (Fig. 1). One important characteristic of the SO is the eastward-flowing Antarctic Circumpolar Current (ACC), which is largely driven by southern westerly winds (SWW) and buoyancy forcing (Rintoul, 2018; Watson et al., 2015). The ACC flows unimpeded around Antarctica and is only slowed down by the South American continent (Orsi et al., 1995), where the northern branch of the ACC bifurcates at $\sim 40\text{–}45°$ S into the northward-flowing Peru–Chile Current (PCC) and the southward-flowing Cape Horn Current (CHC; Strub et al., 1998). CHC and ACC jointly transport ca. 130–150 Sv of water (e.g., Koenig et al., 2014) through the $\sim 800$ km wide Drake Passage into the Atlantic Ocean (Fig. 1).

Several fronts within the ACC characterize the convergence of water masses that differ in temperature, salinity, and nutrient content (Orsi et al., 1995). The northern boundary of the ACC is defined by the subtropical front (STF; Orsi et al., 1995), followed from north to south by the Subantarctic Front (SAF), the polar front (PF), and the southern ACC front (SACCF). Apart from the STF, which is interrupted by the South American continent, all three fronts (SAF, PF, and SACCF) pass through the Drake Passage (Orsi et al., 1995; Fig. 1). The zones between the fronts are defined as areas with differing temperature and salinity characteristics, both decreasing with increasing latitude. The SAF marks the beginning of the northward descent of Antarctic Intermediate Water (AAIW) to a depth of $\sim 500$ m. The AAIW itself is associated with a salinity minimum of $< 34$ PSU. The PF, on the other hand, marks the northern temperature limit of the cold Antarctic surface water. The SACCF instead has no distinct separating features in the surface water. The boundary is here defined along the mesopelagic temperature maximum of the upwelled upper Circumpolar Deep Water (UCDW; Orsi et al., 1995, and references therein).

## 3   Material and methods

A total of 33 multicorer (MUC) samples (Table A2) along the southern Chilean margin and the Drake Passage were analyzed for alkenones and isoGDGTs. The samples were

collected during the R/V *Polarstern* expedition PS97 in February–April 2016 (Lamy, 2016) along a latitudinal transect on the southern Chilean margin and through the Southern Ocean frontal system.

The MUC samples were kept in deep-frozen storage immediately after sampling on board and freeze-dried afterwards in the laboratory. The extraction of the biomarkers was carried out with two different approaches. Between 3 and 5 g of ground surface sediment (0–1 cm) from each site was extracted either by an accelerated solvent extraction (Dionex ASE 350; Thermo Fisher Scientific) with DCM : MeOH (9 : 1; $v/v$) for the samples of the Chilean margin (including three samples from the Drake Passage) or in an ultrasonic bath with DCM : MeOH (2 : 1; $v/v$) for the samples of the Drake Passage. As an internal standard, 100 µL each of the $n$-alkane $C_{36}$ or 2-nonadecanone standard and $C_{46}$ were added before extraction. The two data sets were initially used to address differing research objectives. The samples from the Chilean margin (including three DP samples) were primarily used for extracting alkenones for SST. Previous works on the DP samples, on the other hand, focused on highly branched isoprenoids (HBIs), sterols, and isoGDGTs (Lamping et al., 2021; Vorrath et al., 2020) and were extracted using sonication as a lower recovery of higher unsaturated HBIs is known when using the accelerated solvent extraction (ASE) method (Belt et al., 2014). In contrast, the $TEX_{86}$ index does not appear to be substantially affected by extraction techniques (Schouten et al., 2013b). The good agreement between the three ASE extraction DP samples and the ultrasonic bath samples (Fig. 7d–f) suggests that the two data sets are comparable.

The bulk of the solvent was removed by rotary evaporation, under a nitrogen gas stream or in a Rocket evaporator (Genevac; SP Scientific). The different fractions were chromatographically separated using small glass columns filled with 5 cm of activated silica gel. After adding the sample, the column was rinsed with 5 mL $n$-hexane, 5 or 8 mL $n$-hexane : DCM (1 : 1; $v/v$), 5 mL DCM (dichloromethane), and 4 mL DCM : MeOH (1 : 1; $v/v$) to yield $n$-alkanes, alkenones, and isoGDGTs, respectively. The samples were dried again and transferred into 2 mL vials. For the measurement, the alkenone fractions were diluted with 200–20 µL $n$-hexane, the GDGT fraction was filtered first, and then diluted with 50–120 µL $n$-hexane : isopropanol (99 : 1; $v : v$).

Alkenones were injected with 1 µL solvent and helium as a carrier gas into an Agilent HP 6890 gas chromatograph equipped with a 60 m DB-1ms column and flame ionization detector. The oven temperature was increased from an initial 60 °C up to 150 °C at 20 °C min$^{-1}$ and thereafter by 6 °C min$^{-1}$ to reach 320 °C.

For the GDGT measurements of most DP samples, we refer to the original studies by Lamping et al. (2021) and Vorrath et al. (2020). The other part of the GDGT samples was analyzed on an Agilent 1260 Infinity II ultrahigh-performance liquid chromatography–mass spectrometry (UPLC–MS) system and an Agilent G6125C single quadrupole mass spectrometer. The chromatographic separation was achieved by coupling two UPLC silica columns (Waters Acquity UPLC BEH HILIC; 2.1 × 150 mm; 1.7 µm) and a 2.1 × 5 mm precolumn, as in Hopmans et al. (2016), but with the following chromatographic modifications: mobile phases A and B consisted of $n$-hexane : chloroform (99 : 1; $v/v$) and $n$-hexane : 2-propanol : chloroform (89 : 10 : 1; $v/v/v$), respectively. The flow rate was set to 0.4 mL min$^{-1}$, and the columns were heated to 50 °C, resulting in a maximum backpressure of 425 bar. Sample aliquots of 20 µL were injected with an isocratic elution for 20 min, using 86 % A and 14 % B, followed by a gradient to 30 % A and 70 % B within the next 20 min. After this, the mobile phase was set to 100 % B, and the column was rinsed for 13 min, followed by 7 min re-equilibration time with 86 % A and 14 % B before the next sample analysis. The total runtime was 60 min.

IsoGDGTs were detected using positive ion atmospheric pressure chemical ionization mass spectrometry (APCI-MS) and selective ion monitoring (SIM) of $(M + H)^+$ ions (Schouten et al., 2007) with the following settings: nebulizer pressure 50 psi, vaporizer and drying gas temperature 350 °C, and drying gas flow 5 L min$^{-1}$. The capillary voltage was 4 kV and the corona current $+5$ µA. The detector was set for the following SIM ions: $m/z$ 744 ($C_{46}$ standard), $m/z$ 1302.3 (GDGT-0), $m/z$ 1300.3 (GDGT-1), $m/z$ 1298.3 (GDGT-2), $m/z$ 1296.3 (GDGT-3), and $m/z$ 1292.3 (crenarchaeol and cren' isomer). The resulting scan or dwell time was 66 ms.

## 4    Results and discussion

Our samples are located on a meridional transect along the Chilean margin extending into the DP, with changing environmental conditions, in particular for SSTs, nutrient supply, salinity, and current regimes. The $U_{37}^{K'}$ values range from 0.07 (PS97/079) to 0.38 (PS97/132), with minimum values in the southernmost region and increasing values to the north. All indices from alkenones and isoGDGTs are listed in Table A2. In Sect. 4.1 and 4.2, we compare the two most widely used calibrations for alkenones in this region, namely the subpolar and polar SO Sikes97 calibration and the Müller98 calibration.

### 4.1    Alkenone-based sea surface temperatures

Alkenone-based SSTs calculated with Müller98 range from $\sim 10$ °C in the northernmost locations of our study area to $\sim 1$ °C in the southern part of the Drake Passage to the south of the PF (Fig. 2a, b). SST estimates based on Sikes97 instead range from $\sim 12.5$ °C in the northernmost locations to $\sim 4$ °C in the Drake Passage (Fig. 2c, d). Most values fit closely to the Müller98 calibration line of both the annual mean and summer SSTs but show an offset to the Sikes97 calibration line (Fig. 3). Although Sikes97 was specifically adapted to

the subpolar and polar SO, it generally overestimates modern SSTs in this study area for both the annual mean and summer (Figs. 2 and 3). Our samples for the SE Pacific fit well to Müller98 but not to Sikes97 (Figs. 2 and 3). Because of the overestimation of modern temperatures by Sikes97, we hereafter chose to solely use the Müller98 calibration.

## 4.2 Influence of seasonality on alkenone temperature reconstruction

### 4.2.1 Seasonal signal along the Chilean margin

Our alkenone-based SSTs fit the WOA05-derived annual mean and summer temperatures and show only a small seasonal effect towards warmer SSTs. This observation is also in line with previous data from the northern–central Chilean margin, which yield a slight seasonal effect south of 50° S (Prahl et al., 2006, 2010). Also, a previous study from the Chilean fjord region confirms that the SST signals are only slightly shifted towards summer in the southern Chilean fjord region (Fig. 4; Caniupán et al., 2014). Along the Chilean continental margin, this seasonal summer effect decreases even further to the south to only $\sim 1$ °C (i.e., summer SSTs vs. annual mean) between 50 and 57° S, based on WOA05-derived SSTs (Fig. 4; blue pluses and yellow crosses). This deviation of 1 °C, at least to the north of the SAF, is within the generally accepted error range for alkenone-derived paleo SSTs of $\pm 1.5$ °C (Müller et al., 1998), so that seasonality appears to be negligible here. Only further south in the Drake Passage do deviations of our reconstructed summer temperatures from the annual mean increase to about 2 °C, which are likewise reflected in WOA05-based SSTs (Fig. 4). Model results show a similar trend, with a small deviation from the annual mean of up to 2.5 °C at higher latitudes as well (Conte et al., 2006). Such an increasing poleward seasonality is not unusual, due to a temporal shift of the alkenone production towards summer (e.g., Volkman, 2000). Another effect that could be involved in the increased seasonality is the reduction in the diversity and quantity of coccolithophores through the frontal system of the ACC (e.g., Saavedra-Pellitero et al., 2014, 2019; Vollmar et al., 2022). The coccolithophore assemblages between the PF and the SAF show a significantly reduced diversity compared to north of the SAF. South of the PF, coccolithophorids occur only sporadically and show a reduced diversity (Saavedra-Pellitero et al., 2014). In this region, we are nearing the lower temperature end and thus ecological boundary conditions for coccolithophores. So, alkenone production could be biased towards warmer years within the inevitably large time period of several hundreds of years that comprises the uppermost centimeter of surface sediment.

Not all data can be described by the poleward increase in the seasonal influence, since at two locations along the Chilean margin an annual mean temperature is reflected (Fig. 4; red circles). The first region is located between $\sim 54$–58° S near the Strait of Magellan, where Atlantic waters mix with Pacific waters. The second region encompasses samples in the DP located close to the PF. The PF marks the temperature boundary of the cold Antarctic surface water, which is subducted at the PF and transported northwards (Orsi et al., 1995, and references therein). This vertical water mass structure likely suppresses potential seasonal effects by providing homogenous temperatures throughout the annual cycle, due to a reduced opportunity to build up a warm summer surface layer.

This weakly expressed seasonality in our results, which remains mostly within the error range of Müller98, is in stark contrast to results from other regions, notably the subarctic North Pacific. There, several studies showed a more consistent seasonal shift towards summer and autumn SSTs of 4–6 °C north of the subarctic front, while locations south of the subarctic front reflect an annual mean (Max et al., 2020; Méheust et al., 2013; Prahl et al., 2010). The subarctic front in the North Pacific acts as a natural boundary, creating a highly stratified subarctic surface ocean with a permanent halocline and leading to pronounced seasonal summer warming within strongly stratified surface waters. In contrast, the transition in the South Pacific from subtropical to polar regions is characterized by a lower-salinity gradient and stratification, leading to a less pronounced SAF. The year-round deep mixing within the ACC prevents the formation of a prominent warm water layer during the summer. Thus, subantarctic SSTs would be expected to show less of a seasonal influence on their SST signal.

### 4.2.2 Regional synthesis of seasonality patterns across the South Pacific

We compared the samples from our relatively small study region with published data from the South Pacific Gyre, the central South Pacific, the Aotearoa / New Zealand margin (Jaeschke et al., 2017), and the northern–central Chilean margin (Prahl et al., 2006, 2010), based on the Müller98 and Sikes97 calibrations (Fig. 5). We also calculated the residual temperatures by subtracting the modern WOA05 temperatures at 10 m water depth from our calculated temperatures, which are shown in combination with $U_{37}^{K'}$ against SSTs (Fig. 5). The central South Pacific and Aotearoa / New Zealand margin samples of Jaeschke et al. (2017) spread over a wide area with different conditions. Taking into account a seasonal effect towards summer SSTs, only few samples of this extended data set match with the Sikes97 calibration (Fig. 5a, b; Jaeschke et al., 2017). This is partly in contrast to the work of Jaeschke et al. (2017), who concluded that alkenone-derived SSTs, in general, best reflect the austral summer months when using the Sikes97 calibration. The Müller98 calibration is applicable in the entire extended study area when compared with both annual mean and summer SSTs, reaffirming our decision to use Müller98 for further analyses.

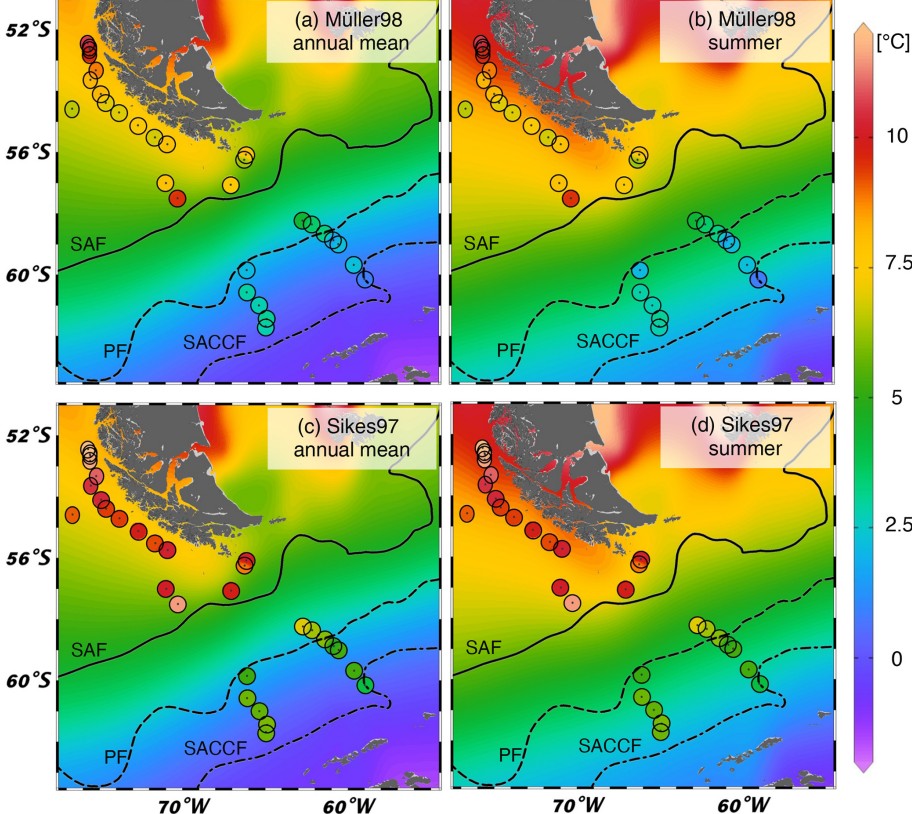

**Figure 2.** Map of reconstructed SST values for $U_{37}^{K'}$ (this study). Background gridded temperatures are derived from WOA05 data (WOA05; Locarnini et al., 2006), and colored dots are calculated SSTs. **(a)** WOA05 annual mean SSTs with Müller98 calibration; **(b)** WOA05 summer SSTs with Müller98 calibration; **(c)** WOA05 annual mean SSTs with Sikes97 calibration; **(d)** WOA05 summer SSTs with Sikes97 calibration.

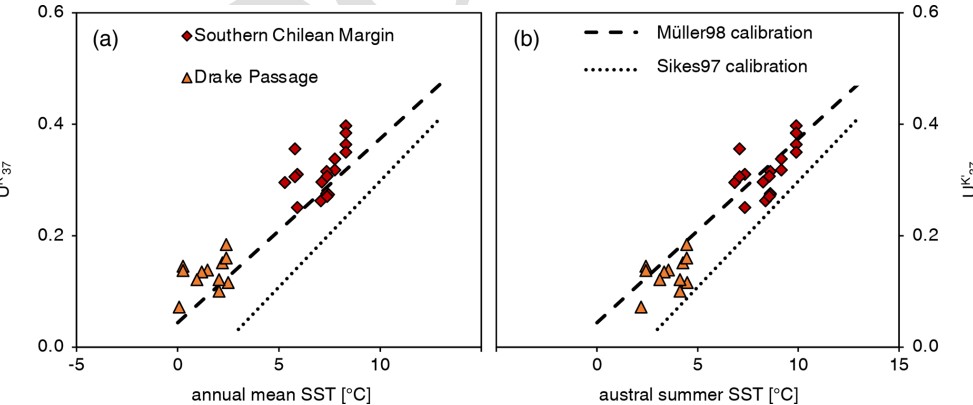

**Figure 3.** Comparison of $U_{37}^{K'}$ index (this study) with modern SSTs at 10 m water depth (WOA05; Locarnini et al., 2006) for **(a)** annual mean SSTs and **(b)** austral summer SSTs, corresponding to January–March.

The SCM and the Drake Passage samples are generally warmer by $\sim 1.5\,°C$ than the samples from the central South Pacific region (Fig. 6). The central South Pacific samples represent a best fit to the annual mean; this is in contrast to the SCM and DP samples, which have a slight, but mostly negligible, seasonal shift toward summer SSTs (Fig. 6). The most likely reason for this bias to summer SSTs in the SE

Pacific could be a higher nutrient availability during the summer months due to the close proximity of the SE Pacific samples to South America. High nutrient availability could lead to a potentially changing competition between different primary producers; e.g., high silica input favors a diatom bloom, which changes when silica is depleted (e.g., Durak et al., 2016; Smith et al., 2017; Tyrrell and Merico, 2004).

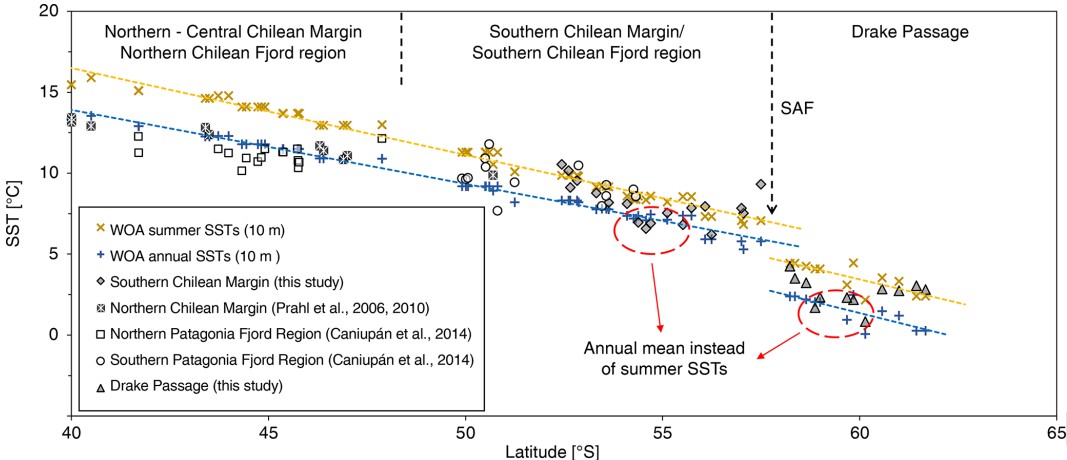

**Figure 4.** Comparison of ocean and fjord samples in the Chilean region. Dashed yellow and blue lines show the meridional temperature evolution during summer and annual mean at 10 m water depth, respectively. Annual mean and summer data were taken from WOA09 (Locarnini et al., 2010) for the samples of Caniupán et al. (2014) and from WOA05 (Locarnini et al., 2006) for Prahl et al. (2006, 2010) and this study.

In this region, nutrient input is expected to be highest during austral summer months, when the high precipitation rates of 3000–10 000 mm yr$^{-1}$ in south Patagonia reach their maximum (e.g., Garreaud et al., 2013; Lamy et al., 2010; Schneider et al., 2003). The increased precipitation during summer months results in an increased freshwater runoff (e.g., Dávila et al., 2002), which is accompanied by an increased supply of continent-derived nutrients to hemipelagic and DP waters and a more stable seasonal thermocline (Toyos et al., 2022) that would both favor a seasonal coccolithophore bloom.

The area off Aotearoa / New Zealand correlates well with samples off the northern–central Chilean margin north of $\sim 45°$ S and corresponds to the annual mean (Fig. 6). In contrast, the South Pacific Gyre samples reflect a summer to autumn signal (Fig. 6; Jaeschke et al., 2017). The South Pacific Gyre is characterized by extremely low nutrient content and an accordingly low primary production (D'hondt et al., 2009). The reasons for the low nutrient content here are the distance from potential continental inputs and a relatively deep thermocline setting, which reduces upwelling and nutrient advection (D'hondt et al., 2009; Lamy et al., 2014). This is reflected by low alkenones, *n*-alkanes, and branched (br)GDGT concentrations (Jaeschke et al., 2017). These factors likely lead to a seasonal bias if, for example, dust transport and macronutrient supply are increased in late spring to summer and relatively quickly exhausted.

## 4.3 GDGT-based (sub)surface temperatures

Similar to $U_{37}^{K'}$ values, isoGDGT-derived indices $\mathrm{TEX}_{86}^H$ and $\mathrm{TEX}_{86}^L$ increase along our transect from south to north. The values range from $-0.48$ to $-0.61$ (PS97/079) and from $-0.39$ to $-0.53$ (PS97/131) for $\mathrm{TEX}_{86}^H$ and $\mathrm{TEX}_{86}^L$, respectively (cf. Table A2). In contrast to coccolithophorids, *Thau-*

*marchaeota* live at greater water depths (e.g., Karner et al., 2001) and also occur in the polar regions (e.g., Massana et al., 1998; Murray et al., 1998), which complicates the choice of an adequate temperature calibration, since reference data and sample sites for both characteristics remain scarce. For the isoGDGTs, we use six calibrations in total for both indices, namely the surface calibrations $\mathrm{SST}^H\mathrm{Kim}$, $\mathrm{SST}^H\mathrm{Kaiser}$, and $\mathrm{SST}^L\mathrm{Kim}$, as well as the subsurface calibrations $T_{\mathrm{sub}}^H\mathrm{Kim}$, $T_{\mathrm{sub}}^H\mathrm{Kaiser}$, and $T_{\mathrm{sub}}^L\mathrm{Kim}$ (Table A1).

Surface and subsurface ranges for isoGDGTs follow the definition of Kaiser et al. (2015) and Kim et al. (2012a, b), with a mean of 0–50 m water depth and 0–200 m water depth, respectively. We therefore used the WOA05-derived temperatures of depths from 10 and 125 m for surface and subsurface, as they roughly correspond to the average values (Fig. 7). Based on the surface calibrations, the temperatures range from $\sim 11.5$ to $5\,°\mathrm{C}$ for $\mathrm{SST}^H\mathrm{Kim}$, from $\sim 9.5$ to $4\,°\mathrm{C}$ for $\mathrm{SST}^H\mathrm{Kaiser}$, and from $\sim 13$ to $6\,°\mathrm{C}$ for $\mathrm{SST}^L\mathrm{Kim}$. With the subsurface calibrations, the temperatures range from $\sim 9$ to $4\,°\mathrm{C}$ for $T_{\mathrm{sub}}^H\mathrm{Kim}$, $\sim 9$ to $5\,°\mathrm{C}$ for $T_{\mathrm{sub}}^H\mathrm{Kaiser}$, and from $\sim 10$ to $5\,°\mathrm{C}$ for $T_{\mathrm{sub}}^L\mathrm{Kim}$ (Fig. 7).

The locations north of the SAF fit best to the modern WOA05-derived SSTs with the $\mathrm{SST}^H\mathrm{Kaiser}$ calibration (Fig. 7b) and appear to extend the surface regression line along the $5\text{--}10\,°\mathrm{C}$ temperature range (Fig. 8a). On average, the modern temperatures are overestimated by $\sim 1.3\,°\mathrm{C}$, which means that they are no longer within the $\pm 0.8\,°\mathrm{C}$ standard error determined by Kaiser et al. (2015) for the surface calibration. In the subsurface, the $T_{\mathrm{sub}}^H\mathrm{Kim}$ and $T_{\mathrm{sub}}^H\mathrm{Kaiser}$ calibrations have an equally good fit with the modern WOA05-derived $T_{\mathrm{sub}}$ (Fig. 7d, e), but the samples tend to fit better with the calibration line of $T_{\mathrm{sub}}^H\mathrm{Kim}$ (Fig. 8b). On average, the modern WOA05-derived $T_{\mathrm{sub}}$ are overestimated here by $\sim 1.6\,°\mathrm{C}$. Thus, the calculated temper-

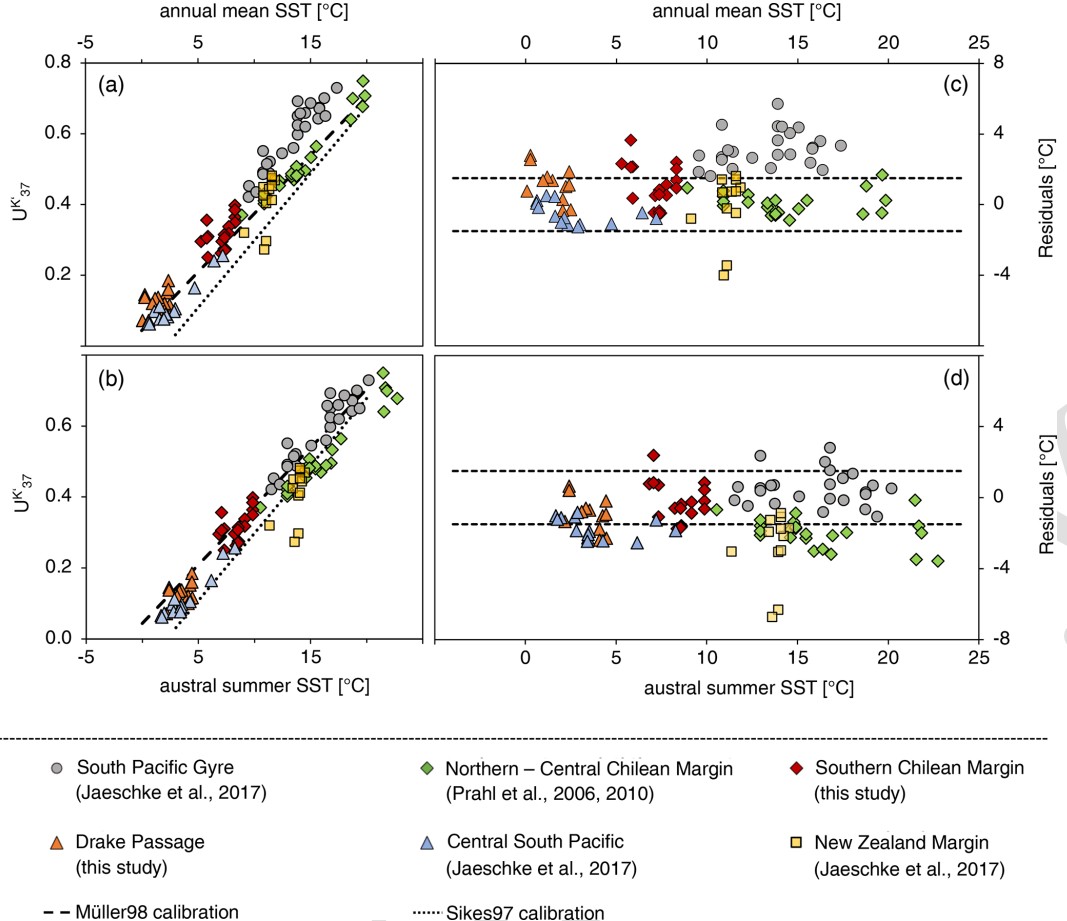

**Figure 5. (a, b)** Compilation of the $U_{37}^{K'}$ index of this study and the expanded South Pacific study area, with SSTs at 10 m water depth for the annual mean and austral summer, respectively (WOA05; Locarnini et al., 2006). **(c, d)** Residuals of the local SSTs at 10 m water depth for the annual mean and austral summer (WOA05; Locarnini et al., 2006) that have been subtracted by the Müller98 calculated SSTs. The temperature range of dotted line shows the standard error in the temperature calibration of $\pm 1.5\,°C$ by Müller et al. (1998). Site PS75/088-6 (Jaeschke et al., 2017) was excluded due to the unrealistic, high temperatures of $> 10\,°C$.

atures are within the $\pm 2.2\,°C$ error range given by Kim et al. (2012a) but not within the $\pm 0.6\,°C$ given by Kaiser et al. (2015) for the subsurface calibration. The samples from the DP, on the other hand, do not fit any calibration and over-estimate the modern WOA05-derived SSTs or $T_{\mathrm{sub}}$ in all calibrations, leading us to compare our results to other previously published data (see Sect. 4.5; Figs. 7 and 8).

Apart from absolute temperature values, the slope of various calibrations allows the calculation of relative temperature changes through time in marine sediment cores. Hence, the slope of the used temperature calibration in an area should adequately resemble the magnitude of relative temperature changes (e.g., between glacial and interglacial periods) to provide correct $\Delta T$, which is often used in modeling studies, for example (Burke et al., 2018), even if the absolute temperature is offset. To determine which calibration best captures the relative temperature changes in our study region, we compared our samples with published data from the central South

Pacific and Aotearoa / New Zealand margin (Ho et al., 2014; Jaeschke et al., 2017) and the northern–central Chilean margin (Kaiser et al., 2015) data set (Fig. 9). In Fig. 9a–d, we show (in red) the regressions of all sites located north of the SAF (hereafter called "local regression") in comparison to the published six different SST and $T_{\mathrm{sub}}$ calibrations of Kim et al. (2010, 2012a and b), and Kaiser et al. (2015). In addition, we show the residuals in Fig. 9e–f to illustrate which data are within the error range of the respective calibrations. For this purpose, the mean WOA05 values of 0–50 and 0–200 m of the annual mean were subtracted from the respective temperature calibration.

The slope of the $\mathrm{SST}^H$ Kim calibration shows a difference of $\sim 3.2$ when compared to our local regression (Fig. 9a) and yields the best relative temperature change across the region north of the SAF, although it generally yields the highest residuals (Fig. 9e). The $T_{\mathrm{sub}}^H$ Kim calibration, with a difference of $\sim 3.9$ between the two slopes (Fig. 9c), captures

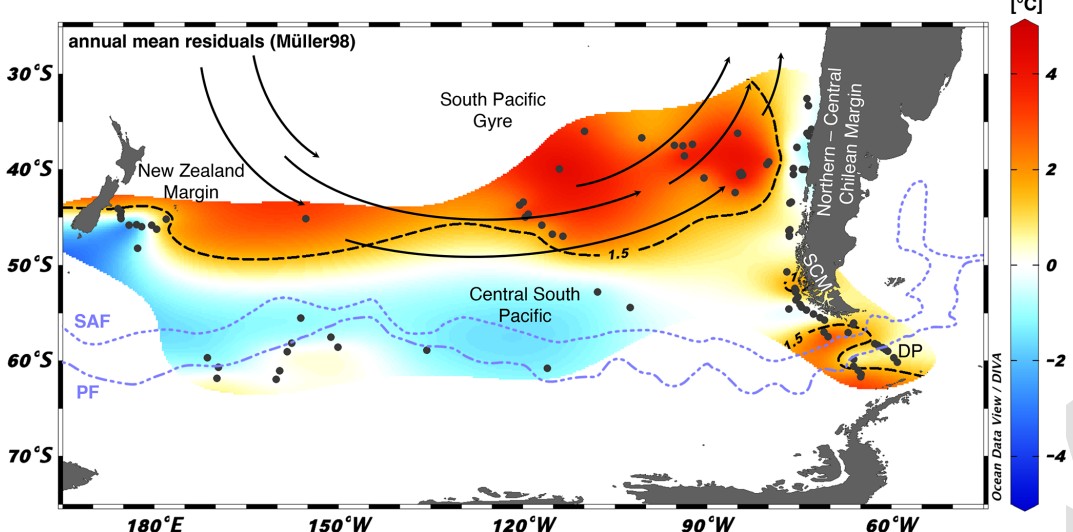

**Figure 6.** Map with residuals for the extended study area of the South Pacific, with published data from the central South Pacific, the Aotearoa / New Zealand margin, the South Pacific Gyre (Jaeschke et al., 2017), and the Chilean margin (Prahl et al., 2006, 2010; this study). Atlas-derived annual mean WOA05 water temperatures of 10 m water depth (Locarnini et al., 2006) were subtracted from the SST Müller98 calibration. SCM is the southern Chilean margin, DP is the Drake Passage, SAF is the Subantarctic Front, and PF is the polar front.

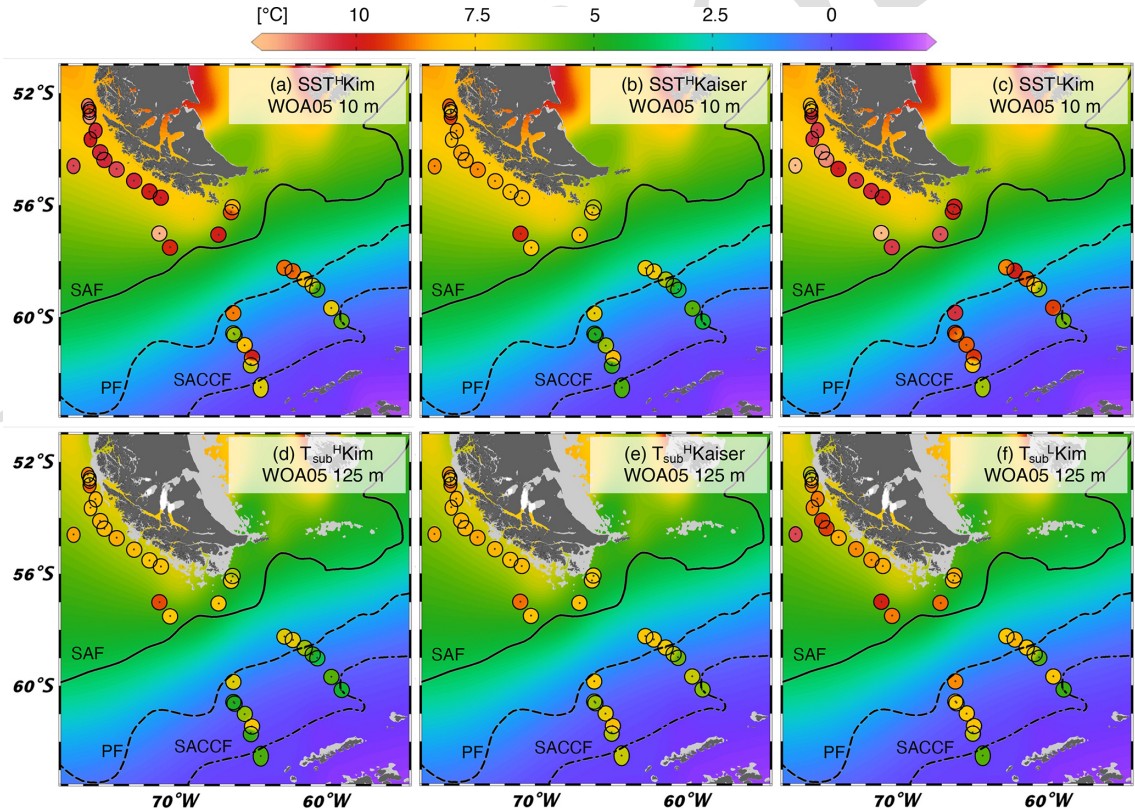

**Figure 7.** Map of reconstructed SST and $T_{sub}$ values for $TEX_{86}^{H}$ and $TEX_{86}^{L}$ (this study). Background gridded annual mean temperatures at 10 or 125 m water depth use WOA05 data, and colored dots are the calculated SSTs or $T_{sub}$. **(a)** WOA05 SSTs with calculated data, using $SST^{H}$Kim. **(b)** WOA05 SSTs with calculated data, using $SST^{H}$Kaiser. **(c)** WOA05 SSTs with calculated data, using $SST^{L}$Kim. **(d)** WOA05 $T_{sub}$ with calculated data, using $T_{sub}^{H}$Kim. **(e)** WOA05 $T_{sub}$ with calculated data, using $T_{sub}^{H}$Kaiser. **(f)** WOA05 $T_{sub}$ with calculated data, using $T_{sub}^{L}$Kim.

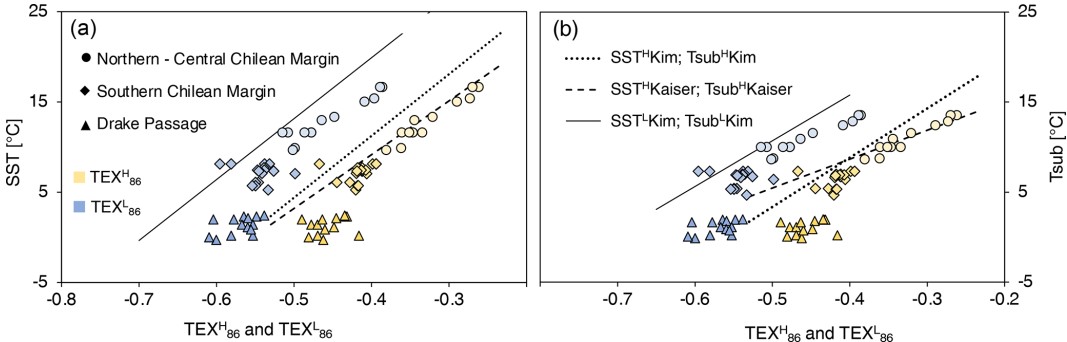

**Figure 8.** Comparison of $TEX_{86}^L$ (blue) and $TEX_{86}^H$ (yellow) data with water (WOA05; Locarnini et al., 2006) with SST 0–50 m water depth (**a**) and $T_{sub}$ 0–200 m (**b**), respectively. The solid black line represents the $TEX_{86}^L$ calibration line ($SST^L$ Kim; $T_{sub}^L$ Kim) for surface and subsurface, respectively. The dashed black line represents the $TEX_{86}^H$ calibration line ($SST^H$ Kaiser; $T_{sub}^H$ Kaiser) for surface and subsurface, respectively. The dotted black line represents the $TEX_{86}^H$ calibration line ($SST^H$ Kim; $T_{sub}^H$ Kim) for surface and subsurface, respectively. Circles show Kaiser et al. (2015) data, the diamonds are related to this study, and the triangles show Lamping et al. (2021) data and this study.

the relative temperature changes as well. The latter corresponds to a temperature change ($TEX_{86}^H$ of $-0.2$ to $-0.3$) of 5.5 °C with $T_{sub}^H$ Kim and 5.1 °C with our local regression. In contrast to $SST^H$ Kim, the residuals are smaller and within the reported error range of $\pm 2.2$ °C (Kim et al., 2012a) for most samples north of the SAF (Fig. 9f). Again, the central South Pacific samples located south of the SAF significantly overestimate local SSTs or $T_{sub}$, with annual residuals of $\sim 8.4$ °C ($SST^H$ Kim), $\sim 6.6$ °C ($SST^H$ Kaiser), $\sim 8.1$ °C ($SST^L$ Kim), $\sim 6.4$ °C ($T_{sub}^H$ Kim), $\sim 6.9$ °C ($T_{sub}^H$ Kaiser), and $\sim 6.7$ °C ($T_{sub}^L$ Kim).

Thus, our combined sample set north of the SAF fits the $T_{sub}^H$ Kim calibration best, while the samples south of the SAF do not match the commonly used calibrations, including the two calibrations based on $TEX_{86}^L$ for (sub)polar regions (Kim et al., 2010, 2012b).

## 4.4 Influence of habitat depth and terrestrial input on *Thaumarchaeota*-derived temperatures

Habitat depth preferences for *Thaumarchaeota* and their response to seasonality (e.g., Schouten et al., 2013a) may influence the $TEX_{86}^H$-derived temperature signals. Since *Thaumarchaeota* are distributed throughout the entire water column, the decision to choose an optimal calibration is closely linked to an initial assumption about the water depth from which the signal originates (Karner et al., 2001). Hence, we applied the ratio of GDGT-2 to GDGT-3 (GDGT [2]/[3]) to locate the water depth of the temperature signal, since subsurface-dwelling *Thaumarchaeota* preferentially yield GDGT-2 over the GDGT-3 (Kim et al., 2015; Taylor et al., 2013).

In the global ocean, the distribution of *Thaumarchaeota* appears to vary within the water column and shows an increasing GDGT [2]/[3] ratio with increasing depth (Dong et al., 2019; Hernández-Sánchez et al., 2014; Kim et al., 2015, 2016; Schouten et al., 2012; Taylor et al., 2013). Water column samples in the Arabian Sea and along the Portuguese margin show a GDGT [2]/[3] ratio between $< 3.3$ in the upper 50 m and $4.0–21.5$ at $> 200$ m water depth (Dong et al., 2019; Kim et al., 2016; Schouten et al., 2012). In the South China Sea, the GDGT [2]/[3] ratio yields $< 3.5$ at $< 100$ m water depth and $5.9–8.6$ at water depth $> 300$ m (Dong et al., 2019). In the southeastern Atlantic, the GDGT [2]/[3] ratio of between 0–50 m water depth is $1.9–3.4$, $4.1–12.8$ between 50–200 m water depth, and $13–50$ in water depth $> 200$ m (Hernández-Sánchez et al., 2014). Thus, increasing GDGT [2]/[3] ratios may not be strictly coupled to water depths across the world ocean. The GDGT [2]/[3] ratio in the surface area seems similar in all regions with $\sim 3.5$, but subsurface values differ considerably. In the southern Atlantic, the GDGT [2]/[3] ratio increases to up to 12.8 within 50–200 m water depth, whereas at the Portuguese margin, the Arabian Sea, and the South China Sea, the GDGT [2]/[3] ratio increases up to $\sim 5.0$, with oxygen content or nutrients being the most likely reason for such non-linearities (e.g., Basse et al., 2014; Villanueva et al., 2015).

The GDGT [2]/[3] ratios in our extended study area vary between $\sim 3–25$. Values $< 5$ ($n = 7$), indicating a surface signal, are found only occasionally off Aotearoa / New Zealand and along the Chilean margin. The majority of samples would correspond to a subsurface signal with a GDGT [2]/[3] ratio $> 5$, confirming our calibration choice ($T_{sub}^H$ Kim) for the South Pacific. Studies from the Peru–Chile Current system, the Antarctic Peninsula, and the North Pacific Gyre confirm this assumption and indicate a subsurface rather than a surface signal (Kalanetra et al., 2009; Karner et al., 2001; Massana et al., 1998; Quiñones et al., 2009). Here, we will distinguish between shallower subsurface (0–200 m water depth) and deep subsurface ($> 200$ m water depth) to

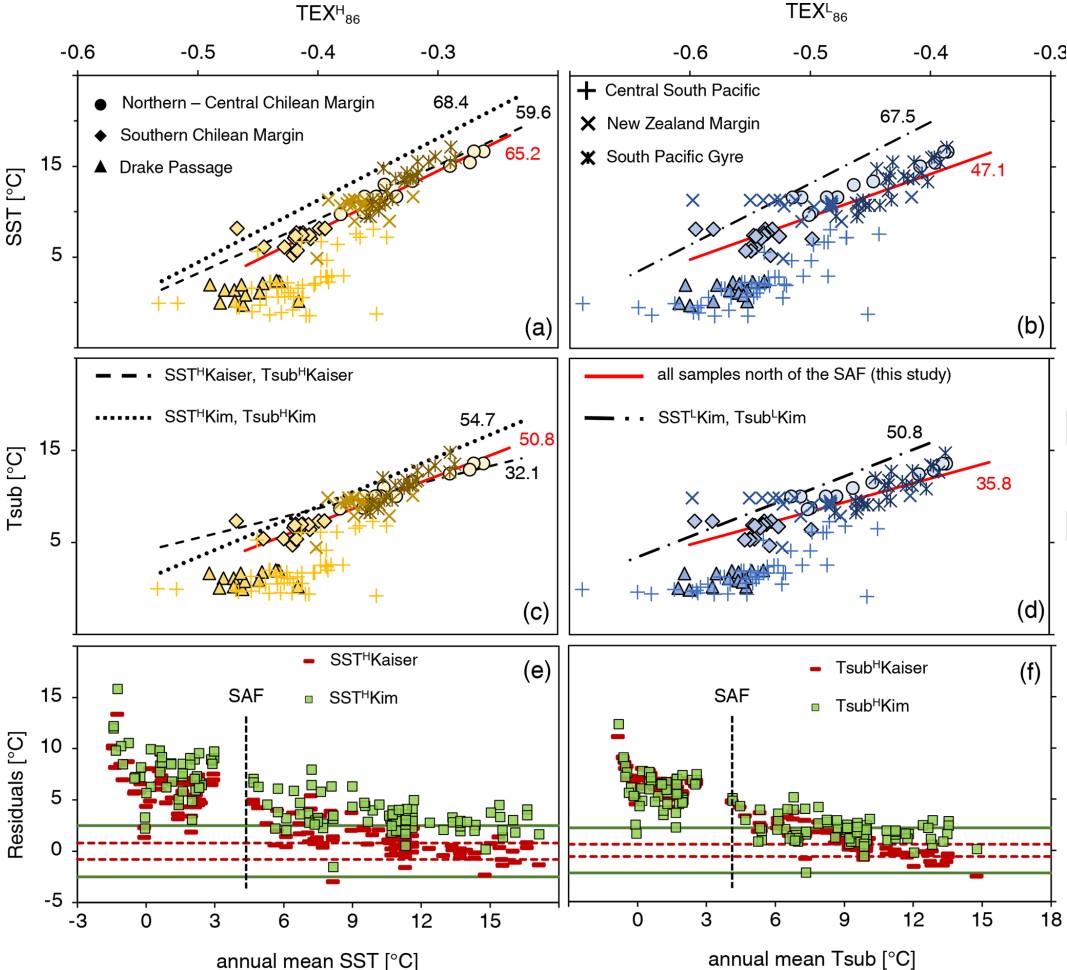

**Figure 9. (a–d)** Comparison of $TEX_{86}^{H}$ (yellow) and $TEX_{86}^{L}$ (blue) data with annual mean water temperature with the SST 0–50 m water depth and $T_{sub}$ 0–200 m (WOA05; Locarnini et al., 2006), respectively. The black (previous studies) and red numbers (this study) indicate the slope of the corresponding calibration. Central South Pacific, Aotearoa / New Zealand margin, and South Pacific Gyre samples from Ho et al. (2014) and Jaeschke et al. (2017) are shown, northern–central Chilean margin samples from Kaiser et al. (2015) are shown, and southern Chilean margin and Drake Passage samples from Lamping et al. (2021) and this study are shown. **(e–f)** Residuals for SST 0–50 m and $T_{sub}$ 0–200 m, with the modern WOA-based temperatures (WOA05; Locarnini et al., 2006) subtracted from the calibrated temperatures. Solid green lines show the standard error of $\pm 2.5\,°C$ ($SST^{H}$Kim) and $\pm 2.2\,°C$ ($T_{sub}^{H}$Kim). Dashed red lines show the calibration standard errors of $\pm 0.8\,°C$ ($SST^{H}$Kaiser) and $\pm 0.6\,°C$ ($T_{sub}^{H}$Kaiser).

quantify the influence of deep subsurface-dwelling *Thaumarchaeota* to the GDGT distribution in the sediment. In general, it is assumed that a deep subsurface water influence is comparatively small, since *Thaumarchaeota* can be most effectively grazed, packed into fecal pellets, and transported to the seafloor within the photic zone (Wuchter et al., 2005). Nevertheless, variations in the GDGT [2]/[3] ratio across the entire study area can provide information about regions that may be subject to a greater influence of deep subsurface-dwelling *Thaumarchaeota*. Our South Pacific locations yield differences in the GDGT [2]/[3] ratio, according to three principally differing boundary or forcing conditions, namely a hemipelagic continental margin setting, a deep thermocline oligotrophic gyre setting, and a SO frontal setting (Fig. 10).

In the overall study area, our results suggest that isoGDGTs record shallower subsurface temperatures rather than surface temperatures. Samples along continental slopes tend to be less influenced by deep subsurface-dwelling *Thaumarchaeota*, while samples from the pelagic regions show a greater influence from deep subsurface-dwelling *Thaumarchaeota*. Samples along the Chilean margin yield a mean GDGT [2]/[3] ratio of $\sim 6.2$ to $\sim 6.9$, reflecting a transition between surface and shallow subsurface habitats. This is in line with northern–central Chilean margin data, which show a positive correlation with both SSTs and $T_{sub}$ (Kaiser et al., 2015). The amount of deep subsurface-dwelling *Thaumarchaeota* increase with increasing distance from land, as shown by samples from the SW Pacific close to

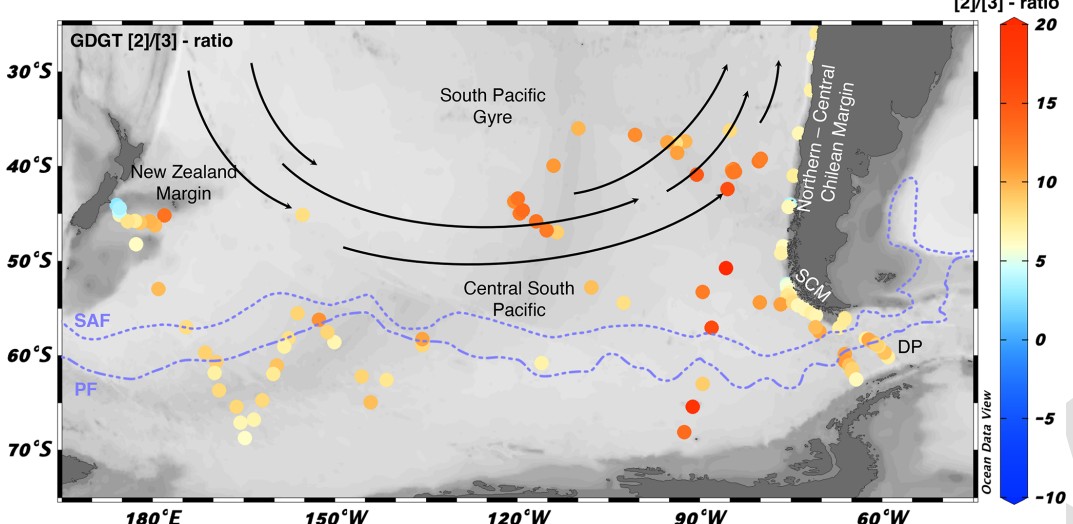

**Figure 10.** Map of the GDGT [2]/[3] ratios from our extended surface sediment sample set across different regions within the study area. SCM is the southern Chilean margin, DP is the Drake Passage, SAF is the Subantarctic Front, and PF is the polar front.

Aotearoa / New Zealand. The GDGT [2]/[3] ratios increase, in line with Taylor et al. (2013), from $\sim 3.1$ at $\sim 600$ m water depth and $\sim 9.8$ at $> 3000$ m water depth to $\sim 12.9$ at $> 4000$ m water depth (Fig. 10). The highest influence of deep-dwelling *Thaumarchaeota* occurs in the South Pacific Gyre and in the eastern South Pacific, averaging a GDGT [2]/[3] ratio of $\sim 11.5$ and indicating a potentially larger contribution of deep subsurface-dwelling *Thaumarchaeota* communities in the sediment, but no significant temperature deviations can be detected for the region north of the SAF (Fig. 9). This suggests that either the influence of the deep subsurface-dwelling *Thaumarchaeota* on the temperature signal is smaller than previously thought or that the distribution of the GDGT [2]/[3] ratio in the subsurface in this region differs from that in the central South Pacific or continental margins.

Besides contributions from deeper living *Thaumarchaeota*, the GDGT [2]/[3] ratio can also be influenced by isoGDGTs derived from terrestrial soils and peats, where the amount of the GDGT-3 is increased compared to the marine milieu (Weijers et al., 2006). This would result in GDGT [2]/[3] ratio decreases with increasing terrestrial input, i.e., in the opposite direction of the influence of deeper living *Thaumarchaeota*. Therefore, we show the GDGT [2]/[3] ratios on a map with monthly average dust depositions (Fig. 12) and found high GDGT [2]/[3] ratios in areas with very little dust accumulation and lower GDGT [2]/[3] ratios with higher dust accumulation, especially during March. This could explain the discrepancy between the eastern and western pelagic South Pacific at a first glance and the fit to the low GDGT [2]/[3] ratios along continental margins. To detect such distorts of terrigenous isoGDGTs on the GDGT [2]/[3] ratio and therefore the TEX-based indices, the

branched vs. isoprenoid tetraether (BIT; should be $< 0.3$) index was developed, where the brGDGTs occurring predominantly in terrestrial soils are related to the crenarchaeol (Hopmans et al., 2004). The BIT is low ($\leq 0.1$) in this region, indicating no significant influence from land (Jaeschke et al., 2017; Kaiser et al., 2015). The visually good correlation between the GDGT [2]/[3] ratios and the dust distribution indicates that future studies could potentially address the underlying causes of these co-variations more systematically, at least in regions with low sedimentation rates but likely high eolian transport. Generally, however, isoGDGT-derived temperatures of the samples north of the SAF fit quite well with $T^H_{\mathrm{sub}}$Kim, so that both potential influences, deeper-dwelling *Thaumarchaeota* and terrigenous input, seem to be negligible.

## 4.5 Towards an alternative Southern Hemisphere (sub)polar calibration for isoGDGT-based temperatures

IsoGDGTs south of the SAF appear to have a lower sensitivity to temperature, which is in line with previous results, showing a large scatter of the $\mathrm{TEX}_{86}$–SST relationship in the polar regions (e.g., Kim et al., 2010; Fietz et al., 2020, and references therein). One reason given for the larger scatter may be a calibration based on satellite-assigned SSTs, which yields values below the freezing point of seawater in polar regions (Pearson and Ingalls, 2013). Consequently, a larger scatter of polar samples leads to a larger error in the estimate in the related calibrations and would explain the occurrence of highest residuals with $\mathrm{SST}^H$Kim and $\mathrm{SST}^L$Kim in our study area in both north and south of the SAF (Figs. 7, 9), where calibrations are based on satellite-assigned SSTs.

However, our data do not show an increased scatter of values south of the SAF. Instead, they show a different $TEX_{86}^H$ ($TEX_{86}^L$)–water temperature relationship, resulting in a lower slope of the calibration line (Fig. 9).

We suspect that the SAF acts as a natural boundary, leading to differential responses within the *Thaumarchaeota* communities and their respective isoGDGTs to changing environmental parameters such as pH (Elling et al., 2015) or oxygen availability (Qin et al., 2015). Another reason for this pattern could be the increased occurrence of OH–isoGDGTs in polar regions. OH–isoGDGTs are present in lower amounts in the sediment than the isoGDGTs used in $TEX_{86}^H$ and $TEX_{86}^L$ but are most abundant in higher latitudes (Fietz et al., 2013; Huguet et al., 2013; Liu et al., 2020). This increased occurrence of OH–isoGDGTs could indicate an adaptation to cold temperatures in order to maintain membrane fluidity. This could simultaneously affect the relationship of TEX-based indices to temperature, thus requiring a separate calibration for high latitudes. OH–isoGDGTs also show a stronger correlation with water temperature than isoGDGTs in both the Arctic (Fietz et al., 2013) and close to Antarctica (Liu et al., 2020), so another OH–isoGDGT-based index of RI-OH' (ring index of hydroxylated tetraethers; equal to $[OH–GDGT-1] + \times [OH–GDGT-2]/[OH–GDGT-0] + [OH–GDGT-1] + [OH–GDGT-2]$; SST is equal to $(RI-OH' - 0.1)/0.0382)$ has been proposed for polar regions (Lü et al., 2015). Moreover, OH–isoGDGTs were often not measured in legacy samples reported in early studies. Based on this, Fietz et al. (2020) recommended a multiproxy approach for the polar regions, which includes both isoGDGTs and OH–isoGDGTs. However, isoGDGTs were more commonly used, and OH–isoGDGTs may not be available for some data sets. A good example is the data sets of, for example, $T_{sub}^L$Kim with $n = 396$ (Kim et al., 2012b) and RI-OH'-based surface calibration with $n = 107$ (Lü et al., 2015), which are designated for the same temperature range $< 15\,°C$. A working TEX-based calibration specifically developed for the SO is therefore appropriate and may also be useful for further research on the functionality of (OH–)isoGDGTs. Therefore, we here take an initial step and propose a modified TEX-based cold temperature calibration for the Southern Hemisphere (sub)polar region. We will then test the new calibration by (1) re-estimating the temperatures of a published sediment core and (2) comparing the SSTs and $T_{sub}$ determined with the new calibration to RI-OH'-based SSTs.

This suggested calibration includes samples south of the SAF in the SO and is extended by the data sets of Kim et al. (2010) and Lamping et al. (2021), with a total of $n = 137$ samples. Changes in $TEX_{86}^H$ or general $TEX_{86}$ indices below $5\,°C$ water temperature (according to SSTs south of the SAF in the Southern Ocean) are less pronounced than above $5\,°C$ because of a weaker correlation of the isomer cren' to water temperatures. Thus, it is excluded in the $TEX_{86}^L$ index, as proposed by earlier studies (Kim et al., 2010). This is in line with

our results in contexts where all $TEX_{86}^L$ calibrations show a stronger correlation than those based on $TEX_{86}^H$ (cf. determination of correlation coefficient $R^2$; Fig. 11). The maximum $R^2$ value of 0.7 ($TEX_{86}^L$; subsurface) is only slightly lower than previously published values ($> 0.8$; cf.Table A1), which is somewhat expected because the scatter of both $TEX_{86}^H$ and $TEX_{86}^L$ indices below $5\,°C$ seems to be generally larger (Ho et al., 2014).

Based on our results, we propose the following new $TEX_{86}^L$-based annual mean, subsurface (0–200 m), and surface (0–50 m) calibration for the Southern Ocean's polar and subpolar regions:

$$T_{sub} = 14.38 \times TEX_{86}^L + 8.93, \qquad (1)$$
$$SST = 10.71 \times TEX_{86}^L + 6.64. \qquad (2)$$

The standard error of $\pm 0.6\,°C$ ($\pm 0.5\,°C$) for our subsurface (surface) calibration is lower than the standard error from the previous subsurface (surface) calibration $T_{sub}^L$Kim ($SST^L$Kim) by $\pm 2.8\,°C$ ($\pm 4\,°C$), which is probably due to the latter's lower data density. With this new calibration, just 44 (40) of the 137 samples lie outside this error range. Another major difference between our calibration and $T_{sub}^L$Kim ($SST^L$Kim) is the slope of the regression line being much flatter here at 14.4 (10.7) than the slope of the calibration $T_{sub}^L$Kim ($SST^L$Kim) of 50.8 (67.5; Fig. 9). This leads to a generally smaller temperature increase with an increasing $TEX_{86}^L$ index. An increase in $TEX_{86}^L$ from $-0.6$ to $-0.5$, for example, corresponds to a temperature change of $1.4\,°C$ in our subsurface calibration and of $5.1\,°C$ in $T_{sub}^L$Kim.

1. Using our new calibration, we compare the recalculated results with the previously used subsurface $TEX_{86}^L$ calibration $T_{sub}^L$Kim (Fig. 13) with core MD03-2601 (66°03.07′ S, 138°33.43′ E; Kim et al., 2012b) from the eastern Indian sector of the SO covering the Holocene, where these authors acknowledged that a temperature offset existed but within the specified calibration error range of $\pm 2.8\,°C$. Temperatures based on our subsurface calibration are on average $\sim 2.5\,°C$ colder than the ones based on the previously used subsurface calibration $T_{sub}^L$Kim. With our new calibration, temperatures remain relatively constant at $-0.8$ and $1.5\,°C$, with the subsurface calibration $T_{sub}^L$Kim between 4.8 and 3.1 ka. Modern temperatures near the core site agree well with the core-top results from our new calibration, with $-0.7\,°C$ (65.5° S, 138.5° E; Locarnini et al., 2006) vs. $-0.8\,°C$ (our reconstruction) for the subsurface layer. The recalculated temperature increases, associated with warmer, nutrient-rich, and modified Circumpolar Deep-Water intrusions, show an attenuated amplitude with the new calibration (Fig. 13). The amplitude based on our new calibration is $1.2\,°C$ around 1.7 ka, whereas it is $4.2\,°C$ with the original subsurface calibration $T_{sub}^L$Kim and therefore a better fit to the expected temperatures.

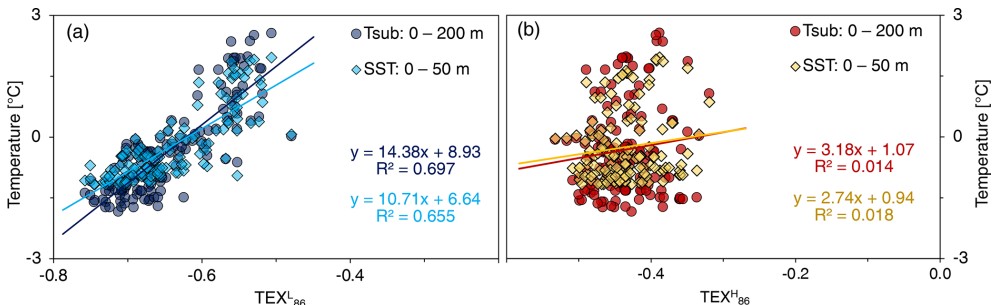

**Figure 11.** $TEX_{86}^{H}$ and $TEX_{86}^{L}$ indices south of the SAF vs. modern WOA05 water temperatures. **(a)** $TEX_{86}^{L}$ of all South Pacific samples. **(b)** $TEX_{86}^{H}$ of all South Pacific samples. TS1

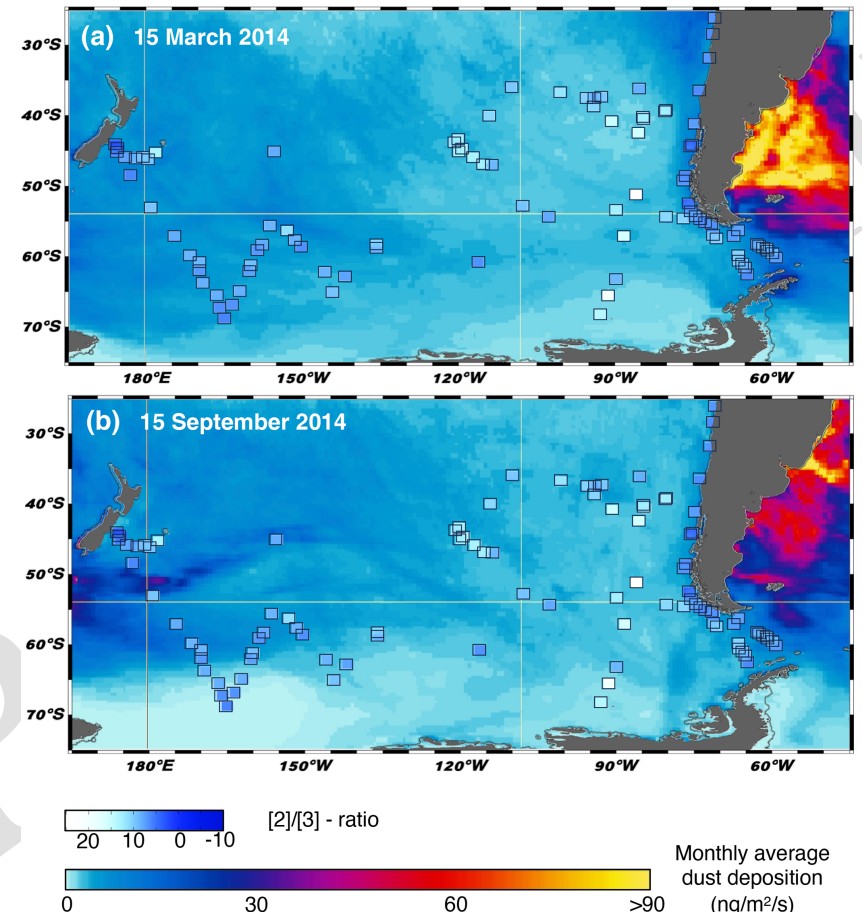

**Figure 12.** GDGT [2]/[3] ratio on a map, showing total monthly average dust deposition (dry + wet) for the times **(a)** March 2014 and **(b)** September 2014. Dust data were taken from NASA Worldview (Global_Modeling_and_Assimilation_Office_(GAMO), 2015).

2. While we do not have OH–isoGDGTs for the Kim et al. (2010), Jaeschke et al. (2017), and Ho et al. (2012) data sets, we calculated SSTs based on the RI-OH' index and compared them to SSTs and $T_{sub}$ based on our calibrations for all samples south of the SAF for which OH–isoGDGTs are available (Drake Passage, Antarctic Peninsula, Weddell Sea, and Amundsen Sea). The results of all three calibrations ($TEX_{86}^{L}$-based SST and

$T_{sub}$; RI-OH'-based SST) fit (Fig. 14), with a temperature discrepancy of ±1 °C from each other. One exception here is the Drake Passage, where the RI-OH'-based SSTs show residuals of > 2 °C. Considering the standard error of ±6 °C (Lü et al., 2015), all RI-OH'-based SSTs are within this error range. The upper-ocean temperatures in the Drake Passage are the only ones in this area above zero, which may explain the larger residuals

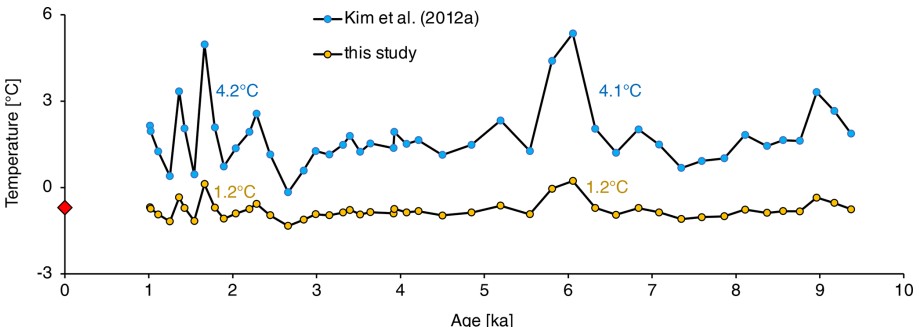

**Figure 13.** Comparison of core MD03-2601 (Kim et al., 2012b) with the temperature calibration $T_{sub}^L$Kim (blue) and the new subsurface calibration of this study (yellow). The red dot marks the mean temperature of 0–200 m water depth at the coring site.

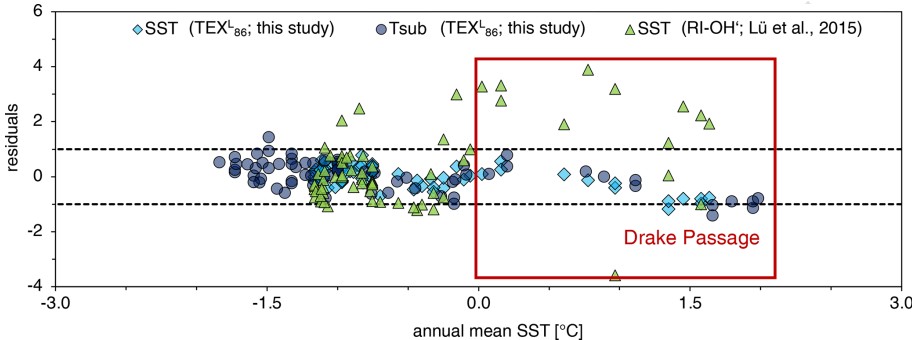

**Figure 14.** Residuals for SST 0–50 m and $T_{sub}$ 0–200 m, with modern WOA-based temperatures (WOA05; Locarnini et al., 2006) subtracted from the calibrated temperatures. Dashed black lines mark the 1 and −1 °C isotherm, respectively.

of the RI-OH'-based SSTs. It is possible that, similar to the subsurface calibration of Kim et al. (2012b), the relative temperature change is not correctly captured due to a regression line that is too steep for samples south of the SAF. The calibration of Lü et al. (2015) includes only a few samples from higher latitudes, most of them from the Arctic, and samples in the temperature range between 1–3 °C are almost absent. A RI-OH'-based calibration for samples south of the SAF could increase the sensitivity of the proxy and decrease the standard error, similar to the $TEX_{86}^L$ index demonstrated here. This is in contrast with the good agreement of the remaining samples, both with our TEX-based temperature reconstructions and with the WOA05-derived temperatures. A calibration based on $TEX_{86}^L$ and developed specifically for the SO yields comparable results to a calibration based on OH–isoGDGTs. We agree with Fietz et al. (2020) that it is useful to determine both indices, if possible, as this is a good way to check the consistency of temperature reconstructions.

## 5 Conclusions

In this study, we provide a qualitative evaluation of the most common temperature calibrations for alkenones and isoGDGTs in the South Pacific and potential environmental influencing factors. For alkenone-derived SSTs, our results provide a best fit with the global core-top calibration of Müller et al. (1998). On a regional scale, the southern Chilean margin and the Drake Passage show a small seasonal effect of $\sim 1$ °C towards warmer SSTs south of $\sim 50°$ S, although well within the $\pm 1.5$ °C standard error for alkenone-derived SSTs (Müller et al., 1998). Excluding local influences, the seasonal effect in the DP is slightly higher at about $\sim 2$ °C and no longer within the error range of calibration by Müller et al. (1998). In contrast, the samples from the central Southern Pacific Ocean show no clear seasonal trend. Causes for this difference between the two areas are the increased seasonal provision of nutrients or more pronounced stratification at the sites proximal to continental runoff along the Chilean margin during the late summertime.

IsoGDGT-based temperatures show a more complex pattern, which necessitates choosing the temperature calibration carefully, depending on the area. The optimal calibration for isoGDGT-based temperature reconstructions in the South Pacific is the subsurface calibration $T_{sub}$ (Kim et al., 2012a) for samples north of the Subantarctic Front, which is in line with evidence from compiled GDGT [2]/[3] ratios that indicate a subsurface of 0 to 200 m water depth, rather than surface habitat depth, throughout the study area. South

of the Subantarctic Front, all existing calibrations overestimate local WOA05-derived temperatures. Furthermore, the GDGT [2]/[3] ratios also correlate with the average monthly dust deposition in the South Pacific. A new calibration for subpolar and polar areas yields lower absolute subsurface temperatures and lower relative changes within the commonly accepted standard error range. The results of this new calibration fit well within the standard error with OH–GDGT-derived temperatures.

For future work, we recommend extending both the geographical area coverage in subpolar and polar regions and the sample density. Furthermore, the influence of seasonality and habitat should be investigated to assess how strongly these factors affect paleo temperature reconstructions TS2.

## Appendix A

**Table A1.** Common indices and their most important temperature calibrations for alkenones and isoGDGTs, with their determination coefficients ($R^2$) and the abbreviations used in this paper. $U_{37}^{K'}$ is the alkenone unsaturation index consisting of 37 carbon atoms; $\mathrm{TEX}_{86}$ is the tetraether index consisting of 86 carbon atoms; $\mathrm{TEX}_{86}^H$ is the tetraether index for water temperatures above 15 °C; $\mathrm{TEX}_{86}^L$ is the tetraether index for water temperatures below 15 °C; SST is the sea surface temperature; and $T_{\mathrm{sub}}$ is the sea subsurface temperature (0–200 m water depth). The abbreviations defined here are used in the main text.

| | Equation | $R^2$ | Abbreviation | References |
|---|---|---|---|---|
| 1 | $U_{37}^{K'} = [\mathrm{C}_{37:2}]/[\mathrm{C}_{37:2}] + [\mathrm{C}_{37:3}]$ | | | Prahl and Wakeham (1987) |
| 2 | $\mathrm{SST} = (U_{37}^{K'} - 0.043)/0.033$ | 0.994 | | Prahl and Wakeham (1987) |
| 3 | $\mathrm{SST} = (U_{37}^{K'} - 0.039)/0.034$ | 0.994 | Prahl88 | Prahl et al. (1988) |
| 4 | $\mathrm{SST} = (U_{37}^{K'} - 0.044)/0.033$ | 0.958 | Müller98 | Müller et al. (1998) |
| 5 | $\mathrm{SST} = (U_{37}^{K'} + 0.082)/0.038$ | 0.921 | Sikes97 | Sikes et al. (1997) |
| 6 | $\mathrm{TEX}_{86} = \dfrac{[2]+[3]+[\mathrm{cren}']}{[1]+[2]+[3]+[\mathrm{cren}']}$ | | | Schouten et al. (2002) |
| 7 | $\mathrm{TEX}_{86}^H = \log \dfrac{[2]+[3]+[\mathrm{cren}']}{[1]+[2]+[3]+[\mathrm{cren}']}$ | | | Kim et al. (2010) |
| 8 | $\mathrm{TEX}_{86}^L = \log \dfrac{[2]}{[1]+[2]+[3]}$ | | | Kim et al. (2010) |
| 9 | $\mathrm{SST} = 68.4 \times \mathrm{TEX}_{86}^H + 38.6$ | 0.87 | $\mathrm{SST}^H$Kim | Kim et al. (2010) |
| 10 | $T_{\mathrm{sub}} = 54.7 \times \mathrm{TEX}_{86}^H + 30.7$ | 0.84 | $T_{\mathrm{sub}}^H$Kim | Kim et al. (2012a) |
| 11 | $\mathrm{SST} = 59.6 \times \mathrm{TEX}_{86}^H + 33$ | 0.91 | $\mathrm{SST}^H$Kaiser | Kaiser et al. (2015) |
| 12 | $T_{\mathrm{sub}} = 32.1 \times \mathrm{TEX}_{86}^H + 21.5$ | 0.86 | $T_{\mathrm{sub}}^H$Kaiser | Kaiser et al. (2015) |
| 13 | $\mathrm{SST} = 67.5 \times \mathrm{TEX}_{86}^L + 46.9$ | 0.86 | $\mathrm{SST}^L$Kim | Kim et al. (2010) |
| 14 | $T_{\mathrm{sub}} = 50.8 \times \mathrm{TEX}_{86}^L + 36.1$ | 0.87 | $T_{\mathrm{sub}}^L$Kim | Kim et al. (2012b) |

**Table A2.** Surface sediment sample results of this study. $U_{37}^{K'}$ is the alkenone unsaturation index consisting of 37 carbon atoms; $TEX_{86}$ is the tetraether index consisting of 86 carbon atoms; $TEX_{86}^{H}$ is the tetraether index for water temperatures above 15 °C; and $TEX_{86}^{L}$ is the tetraether index for water temperatures below 15 °C.

| | Station | Latitude | Longitude | Depth (m) | $U_{37}^{K'}$ | $TEX_{86}^{H}$ | $TEX_{86}^{L}$ |
|---|---|---|---|---|---|---|---|
| Southern Chilean margin | | | | | | | |
| 1 | PS97/139-1 | 52°26.56′ S | 75°42.42′ W | 640 | 0.40 | −0.40 | −0.58 |
| 2 | PS97/134-1 | 52°40.97′ S | 75°34.85′ W | 1075.1 | 0.36 | −0.39 | −0.54 |
| 3 | PS97/132-2 | 52°37.01′ S | 75°35.14′ W | 843 | 0.38 | −0.47 | −0.60 |
| 4 | PS97/131-1 | 52°39.58′ S | 75°33.97′ W | 1028.2 | 0.35 | −0.39 | −0.53 |
| 5 | PS97/129-2 | 53°19.28′ S | 75°12.84′ W | 1879.4 | 0.34 | −0.41 | −0.54 |
| 6 | PS97/128-1 | 53°38.04′ S | 75°32.71′ W | 2293.7 | 0.32 | −0.42 | −0.54 |
| 7 | PS97/122-2 | 54°5.85′ S | 74°54.89′ W | 2560 | 0.31 | −0.41 | −0.53 |
| 8 | PS97/114-1 | 54°34.68′ S | 76°38.85′ W | 3863 | 0.26 | −0.41 | −0.50 |
| 9 | PS97/027-1 | 54°23.05′ S | 74°36.30′ W | 2349.2 | 0.28 | −0.41 | −0.53 |
| 10 | PS97/024-2 | 54°35.27′ S | 73°57.30′ W | 1272.8 | – | – | – |
| 11 | PS97/022-1 | 54°42.03′ S | 73°48.38′ W | 1615.1 | 0.27 | −0.41 | −0.55 |
| 12 | PS97/021-1 | 55°6.91′ S | 72°40.09′ W | 1840.4 | 0.30 | −0.41 | −0.54 |
| 13 | PS97/020-1 | 55°30.80′ S | 71°38.22′ W | 2104.3 | 0.27 | −0.42 | −0.54 |
| 14 | PS97/015-2 | 55°43.89′ S | 70°53.55′ W | 1886.3 | 0.31 | −0.42 | −0.55 |
| 15 | PS97/094-1 | 57°0.17′ S | 70°58.32′ W | 3993.4 | 0.31 | −0.39 | −0.52 |
| 16 | PS97/093-3 | 57°29.92′ S | 70°16.57′ W | 3782.2 | 0.36 | −0.42 | −0.54 |
| 17 | PS97/097-1 | 57°3.27′ S | 67°4.00′ W | 2318.6 | 0.30 | −0.42 | −0.53 |
| 18 | PS97/096-1 | 56°4.53′ S | 66°8.96′ W | 1620.7 | 0.31 | −0.44 | −0.55 |
| 19 | PS97/095-1 | 56°14.68′ S | 66°14.95′ W | 1652.1 | 0.25 | −0.43 | −0.55 |
| Drake Passage–Shackleton Fracture Zone | | | | | | | |
| 20 | PS97/089-2 | 58°13.60′ S | 62°43.63′ W | 3431.9 | 0.18 | −0.43 | −0.56 |
| 21 | PS97/086-2 | 58°38.65′ S | 61°23.82′ W | 2968.9 | 0.15 | −0.45 | −0.56 |
| 22 | PS97/085-2 | 58°21.28′ S | 62°10.07′ W | 3090.7 | 0.16 | −0.43 | −0.55 |
| 23* | PS97/084-2 | 58°52.14′ S | 60°51.91′ W | 3617.4 | 0.10 | −0.46 | −0.58 |
| 24* | PS97/083-1 | 58°59.65′ S | 60°34.28′ W | 3756.3 | 0.12 | −0.49 | −0.60 |
| 25* | PS97/080-2 | 59°40.49′ S | 59°37.86′ W | 3112.7 | 0.12 | −0.46 | −0.56 |
| 26* | PS97/079-1 | 60°8.55′ S | 58°59.42′ W | 3539.3 | 0.07 | −0.48 | −0.61 |
| Drake Passage–Phoenix Antarctic Ridge | | | | | | | |
| 27* | PS97/042-1 | 59°50.62′ S | 66°5.77′ W | 4172 | 0.12 | −0.43 | −0.54 |
| 28* | PS97/044-1 | 60°36.80′ S | 66°1.34′ W | 1202.8 | – | −0.48 | −0.57 |
| 29* | PS97/045-1 | 60°34.27′ S | 66°5.67′ W | 2292 | 0.14 | −0.47 | −0.55 |
| 30* | PS97/046-6 | 60°59.74′ S | 65°21.40′ W | 2802.7 | 0.13 | −0.45 | −0.56 |
| 31* | PS97/048-1 | 61°26.40′ S | 64°53.27′ W | 3455.2 | 0.14 | −0.42 | −0.55 |
| 32* | PS97/049-2 | 61°40.28′ S | 64°57.74′ W | 3752.2 | 0.14 | −0.47 | −0.58 |
| 33* | PS97/052-3 | 62°29.93′ S | 64°17.63′ W | 2889.8 | – | −0.46 | −0.60 |

* isoGDGTs from Lamping et al. (2021); alkenones from this study.

**Data availability.** All locations and the three main indices of the new 33 samples of this study are available in Table A2. All original data will be stored in, and can be accessed through, the PANGAEA Data Publisher repository (https://www.pangaea.de).

**Author contributions.** The study was conceived by JRH and LLJ. MEV, JRH, JH, and NR contributed with analytical tools. JRH, LLJ, JK, and AJ analyzed the data. JRH drafted the paper and prepared the figures. LLJ supervised the study. All authors contributed to the interpretation and discussion of the results and commented on, or contributed to, the draft and final version of the paper.

**Competing interests.** The contact author has declared that none of the authors has any competing interests.

**Disclaimer.** Publisher's note: Copernicus Publications remains neutral with regard to jurisdictional claims in published maps and institutional affiliations.

**Acknowledgements.** We thank the captain and crew of R/V *Polarstern* and the science party for their professional support on expedition PS97 (Paleo–Drake). We thank Sophie Ehrhardt for providing unpublished alkenone data. We thank the technicians Walter Luttmer and Denise Diekstall for their support in the laboratory. We acknowledge the use of imagery from the NASA Worldview application (https://worldview.earthdata.nasa.gov/, last access: 25 May 2023), part of the NASA Earth Observing System Data and Information System (EOSDIS). We thank the two anonymous reviewers and the editor for their constructive, detailed, and very helpful comments, which allowed us to improve our work significantly.

**Financial support.** This research has been supported by the AWI institutional research programs, PACES-II and Changing Earth – Sustaining our Future, and through the REKLIM initiative.

The article processing charges for this open-access publication were covered by the Alfred Wegener Institute, Helmholtz Centre for Polar and Marine Research (AWI).

**Review statement.** This paper was edited by Erin McClymont and reviewed by two anonymous referees.

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

## Remarks from the typesetter