# Peer review of "Upper ocean temperature characteristics in the subantarctic Southeast Pacific based on biomarker reconstructions"

_EGUsphere, 2022_

## Referee Comment (RC1)

Review:

Upper ocean temperature characteristics in the subantarctic Southeast Pacific based on biomarker reconstructions.

Hagemann et al. 2023

Overall comments:

The numbering of lines every 5 lengthens the reviewer task, please number each line next time.

While the article is well written, style is heavy and unnecessarily long, I suggest the authors edit down to go straight to the point.

Authors must add primary-original references or reviews, especially in the introduction.

Please check the following reference for applicability of GDGTs in cold areas:

https://www.jstor.org/stable/26937748

Detailed comments:

**Abstract:**

Writing style should be more to the point-simple sentences that engage the viewer.

Line 19-20: needs clarification residual errors? Sikes overestimates but what is then the residulas?-Please simplify-clarify

Lines 20-23: rearrange-simplify-break into 2 sentences? . Ej: Whereas alkenone estimates  in the Southern Pacific reflect mean annual values.

Line 23: We show that for GDGT-based temperatures, a more complex pattern emerges. This really says nothing, please eliminate and rewrite lines 23-26.

Line 26: Based on a qualitative assessment of the GDGT [2]/[3]-ratios . This is unclear the ratio is a quantitiave value how do you evaluate it qualitatively?

Line 27: 0-200 is not really a subsurface signal since it includes the surface-please clarify

Lines 28-30: Not clear if you are referring to both proxies or only GDGT ones. Also not clear why relative changes would not be reliable if the absolute values are wrong-please explain.

Lines 30-32: Not clear what you did, you suggest a new calibration with TEX86L?-do you suggest TEX86L does not work in the study area?

**Introduction:**

Line 34: missing alkenone reference

Line 35: need to acknowledge review-wider studies for this statement-Herbert reference is only for tropical oceans!

Line 37: this was established way before 2005

Line 43: Please change beginning of sentence, e.g: Henceforth we use….

Line 45-46:missing references

Lines 45-47: I suggest merging into 1-simplify

Line 49-However-you are not contradicting previous statement

Line 50: Unclear what principally wider means-rewrite sentence

Lines 50-54: rewrite:

In the case of our study region, samples from the Central South Pacific are most
likely to represent either summer temperatures (with the Sikes97 calibration) or an annual mean (Muller98 calibration;
Jaeschke et al., 2017). Prahl et al. (2010) instead, using samples from the Chilean continental slope, found a slight seasonal
summer bias south of ~50° S.

Line 54-55:rewrite

Line 57-58: Delete

Lines 59-65: rearrange form general to specific.

Line 70: you need to explain better why it is qualitative

Line 75: I think there are several more articles on this-add artciles or e.g.

Line 76: In the following not how you say this in English

Line 87-89: it seemed in the abstract that the problem was in the northern part?

Line 90-95/100-105: cite fig.1 thought

Lines 112-114: You need to justify the use of two different extraction methods for GDGTs it is important (see Huguet et al. 2010;
https://aslopubs.onlinelibrary.wiley.com/doi/pdfdirect/10.4319/lom.2010.8.127). Were the two extractions used randomly or differential for datasets?

Line 125: you mean extraction or measurement?

Line 126-136: If you used different measuring setups are you sure they give equivalent results? Did you do some cross measurements? This is usually not so straight forward-see intercalibration studies by Schouten et al. 2009-2013. This could potentially bias your results, please elaborate this since it is crucial to validate your results.

**Results and discussion:**

Lines 142-145: I would like a more impactful start of the section-it should makes readers want to continue

Line 144: Again please change- In the following

Section 4.1-Is basically results…Maybe consider separating results form discussion? Or increase the discussion-did other authors compare the two calibrations? Did other studies find similar results?-deviations?

Line 161: Not clear why Fig. 4-different format?

Line 166: you need to frame these results-have others found the same? Why do you think this is the case? Please add discussion.

Lines 167-169: Re-write, simplify, add discussion

Line 171-173: It is not clear why the water column structure will affect signal- seasonality specially since alkenones indicate  SST please clarify.

173-175: equally unclear the effect of diversity on the UK37-Expand

Lines 180-184: re-write, not clear why your results are different form other regions.

Lines 185-190: reads like methods

Lines 190-195: You seem to be discussion Jaeschke's article plaseclarify the link-relation to your results.

Lines 195-200: rewrite, unclear.

Line 199: What do you mean by changing competion?

Lines 204-205: how does stratification favour a bloom?

Lines 210-214: Please rewrite: confusing, organize, simply

Line 219: What you mean by greater latitudes? Are alkenones not globally distributed?

Lines 223-228: I don't understand why you apply TEX86H in this area

Lines 229-232: This makes sense since you apply a calibration inadequate to the area remove or rewrite.

Lines 236-237: Why? Discuss some options-were the lipids extracted-measured in a different way?

Line 239: Time series? Not clear to me

Lines 255-257…:Please discuss why you think this is the case-right now it is only results

SECTION 4.4.: you may want to have a look at the Ingalls et al and Huguet et al. 2022 publications: https://journals.asm.org/doi/full/10.1128/AEM.07016-11

https://www.sciencedirect.com/science/article/pii/S0031018222003091

and references therein to discuss the issue of depth provenance of the signal fully

Line 282: change In the following

Lines 320-324: Please use this to structure and simplify Lines 290-320. Also add some discussion as the reasons why you see changes between different water column-coastal-open ocean settings, otherwise it is just results.

Lines 325-335: Please revise Fietz et al. 2020 (https://www.jstor.org/stable/26937748) publication and references therein. Improve the discussion by better explaining what is the expected effect of NOT including the OH-as seen by others in cold areas.

Lines 367-368: Please clarify that your calibration fits better with actual results.

**Conclusions:**

Lines 371-375: Not conclusions

Line 376: rewrite, strange English

Line 390: You already said what you did, limit yourself to conclusions please.

Line 392: Separate Since in different paragraph-may want to rewrite

**Tables:**

Legends are not self explanatory-please extend.

**Figures:**

Figures are nicely executed, in some cases they seem a bit blurry or text is cut which may be the result of PDF buidlup.

Figure 12 has a frame which I think should be removed.

---

## Author Response (AR1)

**Reply Letter**

for the manuscript "*Upper ocean temperature characteristics in the subantarctic Southeast Pacific based on biomarker reconstructions*" by Hagemann et al. (2023)

Dear Editor and Reviewers,

We are grateful for the Editor´s comments and the constructive reviews that helped us to improve the manuscript. We modified the manuscript accordingly and outline the response to each of the reviewers' comments in the following:

**First Reviewer**

Dear Reviewer #1,

We are grateful for your comments and the constructive review that helped us to improve the manuscript. We modified the manuscript accordingly and outline the response to each of your comments in the following:

**Overall comments:**

**The numbering of lines every 5 lengthens the reviewer task, please number each line next time.**

➔ We are sorry for not numbering all lines in the original version and have rectified this.

**While the article is well written, style is heavy and unnecessarily long, I suggest the authors edit down to go straight to the point.**

➔ We have rewritten several parts of the manuscript, especially in the places where it was recommended by the reviewers to create a better, more straightforward reading flow.

**Authors must add primary-original references or reviews, especially in the introduction. Please check the following reference for applicability of GDGTs in cold areas:**

**https://www.jstor.org/stable/26937748**

➔ We thank the reviewer for bringing additional references to our attention and have added Herbert et al. (2014) and Conte et al. (2006) to the Alkenone-part of the Introduction.

➔ We also added References in the GDGT-part of the Introduction. Some of them are new to the manuscript (Fietz et al. 2020 – as recommended by the reviewer, Fietz et al., 2016; Chong, 2010; Gabriel and Chong, 2000; Schouten et al., 2013), others were already used in subsequent chapters (Fietz et al., 2013; Huguet et al., 2013; Liu et al., 2020). Most of the additional references were added to paragraphs about OH-isoGDGTs (also mentioned from Reviewer 2) as further adaption of Thaumarchaeota to cold areas.

➔ Furthermore, we added a few references in the last chapter of our manuscript: Fietz et al., 2020; Peason & Ingals, 2013; Elling et al., 2015; Qin et al., 2015

**Detailed comments:**

**Abstract:**

**Writing style should be more to the point-simple sentences that engage the viewer.**

➔ We re-wrote the Abstract in parts in order to better outline the main points of the paper.

**Line 19-20: needs clarification residual errors? Sikes overestimates but what is then the residuals? - Please simplify-clarify**

➔ We deleted the term "residuals" out of the abstract and replaced it with "deviation".

**Lines 20-23: rearrange-simplify-break into 2 sentences? E.g.: Whereas alkenone estimates in the Southern Pacific reflect mean annual values.**

➔ We separated the two parts of the sentence into: "We show that for alkenone-derived SSTs, the widely-used global core-top calibration of Müller et al. (1998) yields the smallest deviation of the WOA05-based SSTs. The calibration of Sikes et al. (1997) instead, adapted to higher latitudes and supposed to show summer SSTs, overestimates modern WOA05-based SSTs."

**Line 23: We show that for GDGT-based temperatures, a more complex pattern emerges. This really says nothing, please eliminate and rewrite lines 23-26.**

➔ First part of the sentence has been deleted: "We show that for GDGT-based temperatures the subsurface calibration of Kim et al. (2012a) best reflects temperatures from the WOA05 in areas north of the Subantarctic Front (SAF)."

**Line 26: Based on a qualitative assessment of the GDGT [2]/[3]-ratios. This is unclear the ratio is a quantitative value how do you evaluate it qualitatively?**

➔ We deleted the term qualitatively.

**Line 27: 0-200 is not really a subsurface signal since it includes the surface-please clarify**

➔ Initially, we chose 0 – 50 m for the surface and 50 – 200 m for the subsurface, whereas the calibrations we rely on (Kim et al. 2012 and Kaiser et al., 2015) define subsurface as 0 – 200 m, since Thaumarchaeota occur throughout the water column. While we agree with the reviewer in principle, we chose to adapt the definitions of these previous, widely used works to retain a homogeneous and comparable layout of water depths.

**Lines 28-30: Not clear if you are referring to both proxies or only GDGT ones. Also, not clear why relative changes would not be reliable if the absolute values are wrong-please explain.**

➔ We changed the sentences to: "The overestimation of GDGTs surface and subsurface temperatures south of the SAF highlights the need for a re-assessment of existing calibrations in the polar Southern Ocean. Neither absolute values nor relative changes can be reliably determined with the existing calibrations because the required calibration line has a much lower slope."

**Lines 30-32: Not clear what you did, you suggest a new calibration with TEX86L? - Do you suggest TEX86L does not work in the study area?**

➔ We agree that the term: "which shows a lower temperature sensitivity" is confusing there and deleted it out of the sentence.

➔ The modified version reads: "Therefore, we suggest a modified Southern Ocean $TEX^L_{86}$ – based calibration for surface and subsurface temperatures, which yields principally lower absolute temperatures, aligns more closely with WOA05-derived values, and also with OH-isoGDGT-derived temperatures."

**Introduction:**

**Line 34: missing alkenone reference**

➔ We added two new references: Brassel et al. (1986) and Herbert et al. (2014)

**Line 35: need to acknowledge review-wider studies for this statement-Herbert reference is only for tropical oceans!**

➔ We used Herbert et al. (2010) as an example for the tropics and Sikes et al. (1997) as an example for polar regions. We have now also added Müller et al. (1998) and Conte et al. (2006) as examples for the global ocean.

**Line 37: this was established way before 2005**

➔ We added Brassel et al. (1986).

**Line 43: Please change beginning of sentence, e.g.: Henceforth we use….**

➔ We changed the term "in the following" to Henceforth.

**Line 45-46: missing references; Lines 45-47: I suggest merging into 1-simplify**

➔ We combined two sentences to make clear that the references refer to both statements. We also added Müller et al. (1998): "The accuracy of alkenone-based calibrations can be influenced by other environmental factors besides temperature, such as light levels, changes in growth rate or nutrient availability, but none of these factors seems to have an appreciable effect on the $U^{K'}_{37}$ index (e.g., Caniupán et al., 2014; Epstein et al., 2001; Herbert, 2001; Popp et al., 1998; Müler et al., 1998)."

**Line 49-However-you are not contradicting previous statement**

➔ We deleted "However" accordingly.

**Line 50: Unclear what principally wider means-rewrite sentence**

➔ We rewrote this sentence and deleted "principally": "Seasonality often plays a significant role at high latitudes (e.g., Max et al., 2020; Prahl et al., 2010), due to primary production being more pronounced in the thermal summer season and annual temperature differences that increases with increasing latitude."

**Lines 50-54: rewrite: In the case of our study region, samples from the Central South Pacific are most likely to represent either summer temperatures (with the Sikes97 calibration) or an annual mean (Müller98 calibration; Jaeschke et al., 2017). Prahl et al. (2010) instead, using samples from the Chilean continental slope, found a slight seasonal summer bias south of ~50° S.**

➔ We deleted "In the case" and "instead": "In our study region, samples from the Central South Pacific most likely represent either summer temperatures (with the Sikes97 calibration) or an annual mean (Müller98 calibration; Jaeschke et al., 2017). Prahl et al. (2010), using samples from the Chilean continental slope, found a slight seasonal summer bias south of ~50° S."

**Line 54-55: rewrite**

➔ Previous: "In addition, studies from the North Pacific show that alkenone temperatures in higher latitudes differ from the annual mean by up to 6° C, recording late summer to autumn SSTs instead (e.g., Max et al., 2020; Prahl et al., 2010)."

➔ Now: "In contrast, studies from the North Pacific show a seasonal signal towards late summer to autumn SSTs that differ from the annual mean by up to 6° C (e.g., Max et al., 2020; Prahl et al., 2010)."

**Line 57-58: Delete**

➔ We deleted this sentence

**Lines 59-65: rearrange form general to specific.**

➔ We agree and rearranged this section accordingly: "GDGTs are lipid remains of Thaumarchaeota (formerly called Crearchaeota Group I; Brochier-Armanet et al., 2008) that include a certain number of moieties, which increases with growth temperature (Schouten et al., 2002). The lipids GDGT-0; GDGT-1; GDGT-2; GDGT-3 contain zero to three cyclopentane moieties in their molecule structure, whereas Crenarchaeol and its isomer Cren' feature four cyclopentane and one hexane moieties. These ring structures regulate membrane fluidity of

Thaumarchaeota and change as an adaption to their ambient temperature (Chong, 2010; Gabriel and Chong, 2000; Schouten et al., 2002). "

**Line 70: you need to explain better why it is qualitative**

➜ We changed qualitatively to roughly

**Line 75: I think there are several more articles on this-add articles or e.g.**

➜ We added e.g.,

**Line 76: In the following not how, you say this in English**

➜ We deleted the term "in the following"

**Line 87-89: it seemed in the abstract that the problem was in the northern part?**

➜ No, samples south of the SAF overestimate common TEX-based calibrations by ~14° C. It is written in the Abstract as follows: "We show that for GDGT-based temperatures the subsurface calibration of Kim et al. (2012a) best reflects temperatures from the WOA05 in areas NORTH of the Subantarctic Front (SAF). Temperatures SOUTH of the SAF instead are significantly overestimated by up to 14° C, irrespective of the applied calibration."

**Line 90-95/100-105: cite Fig. 1 thought**

➜ We cited Figure 1 two times more.

**Lines 112-114: You need to justify the use of two different extraction methods for GDGTs it is important (see Huguet et al. 2010; https://aslopubs.onlinelibrary.wiley.com/doi/pdfdirect/10.4319/lom.2010.8.127). Were the two extractions used randomly or differential for datasets?**

➜ The samples for the datasets were originally extracted in order to address different research questions. We added this paragraph as an explanation in the Methods: "Between 3 and 5 g of ground surface sediment (0 – 1 cm) from each site was extracted either by an accelerated Solvent Extraction (DIONEX ASE 350; Thermo Scientific) with DCM:MeOH (9:1, v:v) for the samples of the Chilean margin (including three samples from the Drake Passage) or in an ultrasonic bath with DCM:MeOH (2:1, v:v) for the samples of the Drake Passage. As internal standard, 100 µl each of the n-alkane $C_{36}$ or 2-nonadecanone standard and $C_{46}$ were added before extraction. The two data sets were originally processed and used to answer differing research questions. The samples from the Chilean margin (including three DP samples) were primarily

aimed at extracting alkenones for SST. The DP samples, on the other hand, focused on highly branched isoprenoids (HBIs), sterols and GDGTs (Lamping et al., 2021; Vorrath et al., 2020), and were extracted using sonication, because a lower recovery of higher unsaturated HBIs is known for the use of an ASE (Belt et al., 2014). In contrast, the $TEX_{86}$ index does not appear to be substantially affected by slightly varying extraction techniques (Schouten et al., 2013). The good agreement between the three ASE extraction DP samples and the ultrasonic bath samples (Figure 7 D – F) suggests that the two data sets are comparable."

➔ In the work of Huguet et al. (2010), the ultrasonic bath method yields the highest amounts of GDGTs, however, that paper focuses more on the extraction efficiency of different methods with respect to IPL-GDGTs and not on their influence on the $TEX_{86}$-index.

**Line 125: you mean extraction or measurement?**

➔ We meant measurement and have corrected it.

**Line 126-136: If you used different measuring setups are you sure they give equivalent results? Did you do some cross measurements? This is usually not so straight forward-see intercalibration studies by Schouten et al. 2009-2013. This could potentially bias your results, please elaborate this since it is crucial to validate your results.**

➔ Due to a time lag in the generation of both data sets, two different HPLCs were used. To ensure that no bias is introduced due to the use of different instrumental setups, a laboratory in-house standard was analyzed with the samples. The obtained $TEX^H_{86}$ and $TEX^L_{86}$ values are comparable. Instrument 1: $TEX^H_{86}$ average (n=4) -0.34, SD 0.02; $TEX^L_{86}$ average (n=4) -0.50, SD 0.01. Instrument 2: $TEX^H_{86}$ average (n=4) -0.34, SD 0.01; $TEX^L_{86}$ average (n=4) -0.50, SD 0.02. In addition, samples from the eastern transect in the Drake Passage that have been analyzed by the different instrumental setups match very well (**Fig. 1; Fig. 7 D-E**).

**Results and discussion:**

**Lines 142-145: I would like a more impactful start of the section-it should make readers want to Continue**

➔ We changed the beginning of this chapters: "Our samples are located on a meridional transect along the Chilean margin into the DP, recording changing environmental conditions such as SST, nutrients supply, salinity and current regimes. The $U^{K'}_{37}$ values range …"

**Line 144: Again, please change- In the following**

➔ Changed to "In chapter 4.1 and 4.2"

**Section 4.1 Is basically results... Maybe consider separating results from discussion? ...**

→ We start the alkenone and GDGT chapter briefly with results and linked those to the best calibration. We found a single results chapter to be quite short, as we are only informing the reader of the maximum and minimum values. The choice of calibration for this region, instead, is supposed to be one of the outcomes of the paper and seems to us to be appropriately placed in the discussion.

**... Or increase the discussion-did other authors compare the two calibrations? Did other studies find similar results? Deviations?**

→ We use the data from previous studies of Jaeschke et al. (2017); Ho et al. (2012), Kaiser et al. (2015) and Prahl et al. (2010) for a comprehensive comparison of samples representing the sample region and environmental boundary conditions relevant to our original newly acquired data, labelled under "extended study area". In the following chapters we draw conclusions and provide a synthesis for the entire region, including our new samples. This is discussed in chapter 4.2.2 "Regional synthesis of seasonality patterns across the South Pacific". Jaeschke et al. (2017) also discuss the choice of an optimal alkenone-based calibration. Kaiser et al. (2015) published a regional calibration for the Chilean margin, which we considered in the discussion.

**Line 161: Not clear why Fig. 4-different format?**

→ In this study we focus on the (mostly pelagic) South Pacific, but there are also data of the Chilean fjord region, which support the results. Overall, we wanted to additionally visualize potential latitudinal effects on the SST patterns. In this graphic, we present them to underline the effect of seasonality at the Chilean margin.

**Line 166: you need to frame these results-have others found the same? Why do you think this is the case? Please add discussion.**

→ Regarding the potential reason for the observed pattern, we added a sentence: "Model results show a similar trend, with a small deviation from the annual of up to 2.5° C at higher latitudes as well (Conte et al., 2006). Such an increasing seasonality poleward is not unusual due to a shift of the alkenone production towards summer (e.g., Volkman, 2000)". We have further elaborated on our regional findings in the context of other regions, including data from the Northern Hemisphere (subarctic Pacific, Max et al., 2020), and the pelagic part where no seasonality can be found, in the subsequent part of the discussion (cf. **Chapter 4.2.2.**). Further data in the Drake Passage could not be found to compare.

**Lines 167-169: Re-write, simplify, add discussion**

➔ We changed the sentence to: "In addition, not all data uniformly show a seasonal trend. The poleward increasing seasonal trend along the Chilean margin is discontinuous twice and reflects an annual mean instead (Figure 4; red circles)."

**Line 171-173: It is not clear why the water column structure will affect signal- seasonality, especially since alkenones indicate SST please clarify.**

➔ We changed the sentence: "This vertical water mass structure likely suppresses potential seasonal effects by providing homogenous temperatures throughout the annual cycle, due to a reduced opportunity to build up a warm summer surface layer."

**173-175: equally unclear the effect of diversity on the UK37-Expand**

➔ We moved this textpart up into the preceding paragraph and changed the sentence structure to explain that a decreasing diversity in coccolithophore assemblages, as well as a reduced amount may reflect warmer years, especially as we are at the very edge of their ecological niche.

➔ "Another effect that could be involved in the increased seasonality is the reduction in the diversity and quantity of coccolithophores through the frontal system of the ACC (e.g., Saavedra-Pellitero et al., 2014; Vollmar et al., 2022; Saavedra-Pellitero et al., 2019). The coccolithophore assemblages between the PF and the SAF show a significantly reduced diversity compared to north of the SAF. South of the PF, coccolithophorids occur only sporadically and show a reduced diversity (Saavedra-Pellitero et al., 2014). In this region, we are at the very edge of the ecological niche of coccolithophores, so alkenone production could be biased by warmer years, as the core top sediments contain several 100 years of sedimentation."

**Lines 180-184: re-write, not clear why your results are different from other regions.**

➔ We added a paragraph and explained the differences: "In the South Pacific instead, the year-round deep mixing within the ACC prevents the formation of a prominent warm water layer during the summer. As a result, the former likely leads to pronounced seasonal summer warming within strongly stratified surface waters, whereas the latter less stratified upper ocean would yield less pronounced warming during austral summer months. Thus, subantarctic SSTs would be expected to show less seasonal influence on their SST signal."

**Lines 185-190: reads like methods**

➔ We decided to introduce the definition of residual temperature in this section because it is the first time, we need them and it helps the reader to follow the chapter. We do not find that the definition is better placed in the methods section, since this part is very strictly limited to extraction and measurement. Definitions as well as the selection of the appropriate calibrations for the region are part of the results of this paper and are intentionally listed as such by us.

**Lines 190-195: You seem to be discussion Jaeschke's article please clarify the link-relation to your results.**

➔ We add a sentence: "This is partly in contrast to Jaeschke et al. (2017) who conclude that alkenone-derived SSTs in general best reflect the austral summer months when using Sikes97 calibration."

**Lines 195-200: rewrite, unclear.**

➔ We rewrote this section to: "The SCM and the Drake Passage samples are generally warmer by ~1.5° C than the samples from the Central South Pacific region (Figure 6). The Central South Pacific samples represent a best fit to annual mean, in contrast to the SCM and DP samples, which have a slight, but mostly negligible, seasonal shift toward summer SSTs (Figure 6). The most likely reason for this tendency to summer SSTs in the SE-Pacific could be a higher nutrient availability during the summer months due to the close proximity of the SE Pacific samples to South America. High nutrient availability could lead to a potentially changing competition between different primary producers. For example, a high input of silica favors a diatom bloom, which changes when silica is depleted (e.g., Durak et al., 2016; Smith et al., 2017; Tyrrell and Merico, 2004). In this region, nutrient input is expected to be highest during austral summer months, when the high precipitation rates of 3,000 – 10,000 mm/yr in South Patagonia reach their maximum (e.g., Garreaud et al., 2013; Lamy et al., 2010; Schneider et al., 2003)."

**Line 199: What do you mean by changing competition?**

➔ We add a sentence: "High nutrient availability could lead to a potentially changing competition between different primary producers. For example, a high input of silica favors a diatom bloom, which changes when silica is depleted (e.g., Durak et al., 2016; Smith et al., 2017; Tyrrell and Merico, 2004)."

**Lines 204-205: how does stratification favour a bloom?**

➔ We changed the sentence structure: "The increased precipitation during summer months results in an increased freshwater runoff (e.g., Dávila et al., 2002), accompanied by increased

supply of continent-derived nutrients to hemipelagic and Drake passage waters and a more stable seasonal thermocline (Toyos et al., 2022), which would both favor a seasonal coccolithophore bloom."

**Lines 210-214: Please rewrite: confusing, organize, simply**

➔ We reorganized this paragraph: "The South Pacific Gyre is characterized by extremely low nutrient content and accordingly low primary production (D'hondt et al., 2009). Reasons for the low nutrient content here are the distance from potential continental inputs, and a relatively deep thermocline setting, which reduces upwelling and nutrient advection (D'hondt et al., 2009; Lamy et al., 2014). This is reflected by low alkenones, n-alkanes and brGDGTs concentrations (Jaeschke et al., 2017)."

**Line 219: What you mean by greater latitudes? Are alkenones not globally distributed?**

➔ While coccolithophores are indeed nearly globally distributed, they decrease significantly (i.e., nearly exponentially) in abundance in subpolar and polar waters. So, while a few studies have found coccolithophores even south of 60° S (e.g., Winter et al., 1999), their abundance is often too low/insignificant to be used for biomarker studies/SST reconstructions. To clarify this, we add the term "also in the polar regions" to the sentence structure.

➔ New Reference: Winter, A., Elbrächter, M. & Krause, G. Subtropical coccolithophores in the Weddell Sea. Deep Sea Research Part I: Oceanographic Research Papers 46, 439–449 (1999).

**Lines 223-228: I don't understand why you apply TEX86H in this area**

➔ North of the SAF (at least at the Chilean continental margin) the GDGTs isomer Crenarchaeol' is present in sufficient quantities and $TEX^H_{86}$ based calibrations show better matches to WOA-derived SST than $TEX^L_{86}$-based calibrations. Since $TEX^H_{86}$ is only a logarithmic function of $TEX_{86}$, we checked only one of these indices for simplicity. Also, there was a local calibration (Kaiser et al., 2015) that we wanted to include in the discussion as well.

**Lines 229-232: This makes sense since you apply a calibration inadequate to the area remove or rewrite.**

➔ We decided to include all main existing $TEX^H_{86}$- and $TEX^L_{86}$-based calibrations for this study area in order to provide a comprehensive overview.

**Lines 236-237: Why? Discuss some options-were the lipids extracted-measured in a different way?**

➔ We have added to the sentence and refer to the next chapter, where we discuss this fact, taking into account an extended data set. "The samples from the DP instead do not fit to any calibration and overestimate modern WOA05-derived SSTs or Tsub in all calibrations, leading us to compare our results to other previously published data (see **Chapter 4.5**; **Figure 7** and **Figure 8**)."

**Line 239: Time series? Not clear to me**

➔ We thank the reviewer and have added that we (i) mean time series based on sediment core reconstructions of temperatures through time, and (ii) relative changes to be often useful for comparison e.g., with other studies, so the slope of the regression is also important.

**Lines 255-257…: Please discuss why you think this is the case-right now it is only results**

➔ The first part of the alkenone and the GDGT chapter is supposed to present shortly the results. Based on them we checked the best calibration fit for the region.

**SECTION 4.4.: you may want to have a look at the Ingalls et al and Huguet et al. 2022 publications:**

**https://journals.asm.org/doi/full/10.1128/AEM.07016-11**

**https://www.sciencedirect.com/science/article/pii/S0031018222003091**

**and references therein to discuss the issue of depth provenance of the signal fully**

➔ We agree with the reviewer and have added a paragraph of the potential input from land, even in a pelagic setting, which has an opposite effect to the depth discussion before.

➔ We also added for this argumentation a new Figure (12)

➔ "Apart from the contribution of deeper living Thaumarchaeota, the GDGT [2]/[3]-ratio can also be influenced by isoGDGTs derived in terrestrial soils and peats, where the amount of the GDGT-3 is increased compared to the marine milieu (Weijers et al., 2006b). Meaning that in the opposite to the influence of deeper living Thaumarchaeota, the GDGT [2]/[3]-ratio decreases with increasing terrestrial input. Therefore, we compared the GDGT [2]/[3]-ratios with monthly average dust depositions (Figure 12) and found high GDGT [2]/[3]-ratios in areas with very little dust entrance and lower GDGT [2]/[3]-ratios with higher dust entrance. This could explain the discrepancy between eastern and western pelagic South Pacific at a first glance and fit also to the low GDGT [2]/[3]-ratios at the continental margins. To detect such distortions of terrigenous isoGDGTs on the GDGT [2]/[3]-ratio and therefore the TEX-based indices, the branched vs isoprenoid tetraether (BIT; should be <0.3) index was developed, where the predominantly in terrestrial soils occurring brGDGTs are related to the Crenarchaeol (Hofman et al., 2004). The BIT is low (≤0.1) in this region, indicating no significant influence from land

(Jaeschke et al., 2017; Kaiser et al., 2015). We recommend due to the perfect fit of the GDGT [2]/[3]-ratios with the dust distribution more studies in the future should be considered, at least in regions with low sedimentation rates but high potential eolian transport. But all in all, isoGDGT-derived temperatures of samples north of the SAF fit quite well with Tsub[H]Kim, so that both potential influences, deeper-dwelling Thaumarchaeota as well as terrigenous input seems to be negligible."

**Line 282: change In the following**

➔ We changed the term "In the following" to "Here"

**Lines 320-324: Please use this to structure and simplify Lines 290-320. Also add some discussion as the reasons why you see changes between different water column-coastal-open ocean settings, otherwise it is just results.**

➔ We have shortened this section by half a page to: "In the overall study area, our results suggest that GDGTs record shallower subsurface temperatures rather than surface temperatures. Samples along continental slopes tend to be less influenced by deep subsurface-dwelling Thaumarchaeota, while samples from the pelagic regions show a greater influence by deep subsurface-dwelling Thaumarchaeota. Samples along the Chilean Margin yield a mean GDGT [2]/[3]-ratio of ~6.2 to ~6.9, reflecting a transition between surface and shallow subsurface habitats. This is in line with Northern – Central Chilean Margin data, which show a positive correlation with both SSTs and Tsub (Kaiser et al., 2015). The amount of deep subsurface-dwelling Thaumarchaeota increase with increasing distance from land, as shown by samples from the SW Pacific close to New Zealand. The GDGT [2]/[3]-ratios increase, in line with Taylor et al. (2013), from ~3.1 at ~600 m water depth to ~9.8 at >3000 m water depth to ~12.9 at >4000 m water depth (**Figure 10**). The highest influence of deep-dwelling Thaumarchaeota occurs in the South Pacific Gyre, averaging a GDGT [2]/[3]-ratio of ~11.5 and indicating a potentially larger contribution of deep subsurface-dwelling Thaumarchaeota communities in the sediment. But it should be mentioned that no significant temperature deviations can be detected for this region (**Figure 9**). This suggests either that the influence of the deep subsurface-dwelling Thaumarchaeota on the temperature signal is smaller than previously thought, or that the distribution of the GDGT [2]/[3]-ratio in the subsurface in this region differs from that in the central South Pacific or continental margins."

**Lines 325-335: Please revise Fietz et al. 2020 (https://www.jstor.org/stable/26937748) publication and references therein. Improve the discussion by better explaining what is the expected effect of**

**NOT including the OH-as seen by others in cold areas.**

→ We revised Fietz et al. (2020) and added this reference together with some more to the chapter, to improve the discussion.

→ We added a paragraph in the last chapter where we show that that our new calibration and the OH-RI' based calibration show, with one exception, comparable results. It seems that a SO TEX$^L_{86}$-based calibration works as well as a RI-OH' calibration.

**Lines 367-368: Please clarify that your calibration fits better with actual results.**

→ We have added the following to the sentence: "…and therefore fits better to the expected temperatures."

**Conclusions:**

**Lines 371-375: Not conclusions**

→ We shortened this paragraph to the sentence: "With this work, we provide a qualitative evaluation of the most common temperature calibrations for alkenones and isoGDGTs in the South Pacific and potential environmental influences on them."

**Line 376: rewrite, strange English**

→ Rewritten. "Our results provide for alkenone-derived SSTs a best fit with the global core-top calibration of Müller et al. (1998)".

**Line 390: You already said what you did, limit yourself to conclusions please.**

→ We deleted the first sentence to avoid repetitions.

**Line 392: Separate Since in different paragraph-may want to rewrite**

→ We separated this Lines and rewrote this paragraph slightly: "For future work, we recommend to extend both the geographical area coverage in subpolar and polar regions and the sample density. Furthermore, the influence of seasonality and habitat should be investigated to assess how strongly these factors affect paleo-temperature reconstructions."

**Tables:**

**Legends are not self explanatory-please extend.**

→ We added an index explanation

**Figures:**

**Figures are nicely executed; in some cases, they seem a bit blurry or text is cut which may be the result of PDF build up.**

→ Yes, it's the PDF

**Figure 12 has a frame which I think should be removed.**

→ Frame is also due to the PDF, will be removed.

**Second Reviewer**

"Upper ocean temperature characteristics in the subantarctic Southeast Pacific based on biomarker reconstructions" by Julia Rieke Hagemann et al.

Dear Reviewer #2,

We sincerely thank you for your comments and the constructive review that helped us to improve the manuscript. We modified the manuscript according to your suggestions and outline the response to each of your comments in the following. With regard to reviewer #1's additional suggestions, which you alluded to, we would kindly refer you also to our respective author comments.

**The authors present a new dataset of alkenone- and GDGT-derived temperatures from the Southern Chilean Margin and the Drake Passage. They evaluate different existing temperature-calibrations and some reasons for bias in derived-temperatures.**

**They have also included other surface sediment samples previously published from the Central and Western South Pacific towards the New Zealand continental margin. Therefore, the authors have made a nice work including several surface sediments to well cover the study area, including samples more influenced by Polar waters and other by Subtropical waters.**

**The idea and methods are well, with the most important implication about the difficulties in the use of different paleothermometers at high latitudes marine areas. In general, I find the manuscript well written, and figures are appropriated.**

**I recommend publishing this article after some comments and revisions will be addressed. Some of them have already been pointed out by the Reviewer #1.**

➔ We thank the reviewer for his/her positive evaluation of our work.

**One concern is about OH-GDGTs. Authors describe the major abundance of OH-GDGTs at high latitudes and their used as paleothermometer, but it seems they have not measured the OH-GDGTs. I wonder if they can include these data and compare how are derived temperatures using RI-OH' and the proposed TEXL86 0-200 m specific calibration.**

**In any case, I would include in the Abstract and Conclusions some sentences encouraging the measurement of OH-GDGTs in the coming GDGTs analysis in order to increase the dataset to improve future calibration works, either for RI-OH' or TEX86 indices.**

➔ We added a paragraph with additional information's about RI-OH'. We also compared our TEX$^L_{86}$-based SSTs and Tsub with the RI-OH'-based SST for all samples we have. We also made a new Graphic (Figure 14) to present these results.

**Minor revisions:**

**Line 45: alkenone-based**

➔ We have corrected the wording

**Line 49: at high latitudes**

➔ We agree and changed according to the recommendation.

**Line 119: Do you add the internal standards before extraction or before column chromatography separation?**

➔ We moved the sentence to the extraction part.

**Line 120-121: Did you filter the GDGTs fraction?**
**In order to be consistent with the rest, I would indicate that the solvent of n-hexane:isopropanol was (99:1, v:v).**

➔ We agree and corrected it.

**Line 125: Please, as comment Reviewer#1, specify why there was different extractions method used and GDGTs measurements.**

➔ We have supplemented this in detail in the methods section. "Between 3 and 5 g of ground surface sediment (0 – 1 cm) from each site was extracted either by an accelerated Solvent Extraction (DIONEX ASE 350; Thermo Scientific) with DCM:MeOH (9:1, v:v) for the samples of the Chilean margin (including three samples from the Drake Passage) or in an ultrasonic bath with DCM:MeOH (2:1, v:v) for the samples of the Drake Passage. As internal standard, 100 µl each of the n-alkane $C_{36}$ or 2-nonadecanone standard and $C_{46}$ were added before extraction. The two data sets were used to answer different research questions. The samples from the Chilean margin (including three DP samples) were primarily aimed at extracting alkenones for SST. The DP samples, on the other hand, focused on highly branched isoprenoids (HBIs), sterols and GDGTs (Lamping et al., 2021; Vorrath et al., 2020), and were extracted using sonication as a lower recovery of higher unsaturated HBIs is noted for using the ASE (Belt et al., 2014). In contrast, the TEX$_{86}$ index does not appear to be substantially affected by extraction techniques (Schouten et al., 2013). The good agreement between the three ASE extraction DP samples

and the ultrasonic bath samples (Figure 7 D – F) suggests that the two data sets are comparable."

**Lines 161 and 192: Figure instead Fig., to be consistent with the rest of the manuscript.**

➔ Thank you for the observation, we have corrected it.

---

## Author Response (AR2)

**Reply Letter2**

for the manuscript "*Upper ocean temperature characteristics in the subantarctic Southeast Pacific based on biomarker reconstructions*" by Hagemann et al. (2023)

Dear Editor and Reviewers,

Thank you for your nice review and please find attached our corrected manuscript.

**Public justification (visible to the public if the article is accepted and published)**:

Thank you for your comprehensive replies to the two reviewers, and your submission of your tracked changes document. For the most part the concerns raised by the reviewers have been addressed. There are some minor typos or corrections which will help to further clarify the text:

**Check author names (e.g., Brassell not Brassel, Muller not Muler...) and spellings (e.g., Weddell not Wedell).**

→ We read the paper carefully and tried to correct smaller mistakes, e.g., "Wedell Sea"

→ Furthermore, we also checked over the above-mentioned citations.

**New text in reply to lines 167-169 (reviewer 1): "In addition, not all data uniformly show a seasonal trend. The poleward increasing seasonal trend along the Chilean margin is discontinuous twice and reflects an annual mean instead (Figure 4; red circles)." This is still confusing, and could perhaps be re-phrased: "Not all data can be described by the poleward increase in the seasonal influence, since at two locations along the Chilean margin an annual mean temperature is reflected instead"**

→ We agree with the Editor and have used the suggested wording.

**Reply to reviewer 1 lines 180-184: Does the second sentence contradict the first? The "latter" of sentence 2 sounds like it is referring to the formation of a prominent mixed layer, yet the text suggests that this is referring to a "less stratified upper ocean signal" if you use only the sentences here in isolation. I can see from your tracked manuscript that in fact you are referring to processes in earlier sentences. Instead of referring to former/latter can you be specific about what you are referring to? e.g. "In the South Pacific, the year round deep mixing within the ACC prevents the formation of stratified waters and a prominent warm water**

**layer during the summer, so that subantarctic SSTs are expected to show less seasonal influence on their signal than the North Pacific"**

➔ We agree with the reviewer and moved part of the sentence "... and to pronounced seasonal summer warming within strongly stratified surface waters." to the sentence part that addresses the north pacific.

➔ The total paragraph mow reads: "The subarctic front in the North Pacific acts as a natural boundary, creating a highly stratified subarctic surface ocean with a permanent halocline and to pronounced seasonal summer warming within strongly stratified surface waters. In contrast, the transition in the South Pacific from subtropical to polar regions is characterized by a lower salinity gradient and stratification, leading to a less pronounced SAF. The year-round deep mixing within the ACC prevents the formation of a prominent warm water layer during the summer. Thus, subantarctic SSTs would be expected to show less seasonal influence on their SST signal."

**In your reply to reviewer 1 Section 4.4 you refer to the relationship between GDGT [2]/[3] ratio and dust, and refer to Figure 12 and a "near-perfect relationship". I find it very difficult to confirm such a strong relationship with this Figure, partly because the colour palette is almost but not completely the same for the underlying dust and the GDGT ratio. For a "near perfect" fit I'd expect to see e.g. oranges and reds for high [2]/[3] overlying oranges and reds for low dust, whereas low dust is blue and on a different colour spectrum to the GDGTs (yellow for dust is max, whereas yellow for GDGTs seems to be intermediate?). Either use scatter plots for extracted data (as you do earlier) or try to align the colour scales for both data sets. At present Figure 12 does not support the strongly worded text.**

➔ We agree with the Editor and changed the wording from "near perfect fit" to "visually good correlation"

➔ Furthermore, we have reworked our graphic and adjusted the color scale for the GDGT 2/3-ratio, to better emphasize the correspondence. The color scale now ranges from white (high GDGT 2/3-ratio) to dark blue (low GDGT 2/3-ratio). We have not considered the yellow color spectrum of the dust distribution, because a) we have no data in the region east of South America and b) it is a "relative" correlation, based on scale I set on e.g., the GDGT 2/3-ratio. The graph is only meant to demonstrate that in the South

Pacific, the dust influence on the TEX-index might play a role as motivation for future works.

**Line 76 in the tracked manuscript includes some leftover text from the previous version ("The number of moieties")**

➔ We thank the editor for mentioning this and deleted this term.

**Line 223 in the tracked manuscript needs WOA for world ocean atlas**

➔ We corrected it.

**Notification to the authors by Polina Shvedko:**

Please ensure that the colour schemes used in your maps and charts allow readers with colour vision deficiencies to correctly interpret your findings. Please check your figures using the Coblis – Color Blindness Simulator (https://www.color-blindness.com/coblis-color-blindness-simulator/) and revise the colour schemes accordingly.

➔ We thank Polina Shvedko for this comment and checked our Figures with the website. Most of them are fine for Red/Green/Blue-Blind people. It can be difficult for Red/Green-blind people to recognize the warmer temperatures in the maps, but since the focus of this work is in the southern area, we think this is fine. We have to mention here that nearly all maps are difficult to read for Monochromacy/Achromatopsia people. Furthermore, in most of the Graphics are different symbols used together with colors, which simplifies the recognition of the essential statements of the figures. All in all, we think the figures are quite fine also for color blind people.